# NEURAL WEIGHT COMPRESSION FOR LANGUAGE MODELS

## ABSTRACT

The efficient storage and transmission of *language model weights* is becoming increasingly important, as their scale and adoption continue to grow. However, as our understanding of this new data modality is limited, designing a good compression algorithm for language model weights heavily relies on manual, trial-and-error approaches. In this paper, we propose a learned compression framework that trains neural codecs directly from pretrained language model weights. Unlike conventional data (*e.g.*, images), language model weights pose unique challenges: the sizes and shapes of weight tensors vary significantly, and the reconstruction quality must be judged by downstream model predictions rather than naïve MSE loss. To address this, we introduce Neural Weight Compression (NWC), a novel autoencoder-based neural codec tailored to model weight compression. The proposed method inherits the advantages of autoencoder-based codecs while incorporating three technical components: (1) column-wise tensor chunking and normalization; (2) an importance-aware training loss; (3) an inference-time error compensation mechanism guided by model outputs. Experiments on open-weight language models show that NWC achieves competitive or state-of-the-art accuracy-compression tradeoffs, with particularly strong results at 4–6 bit precisions where accuracy remains nearly on par with FP16 models.

## 1 INTRODUCTION

The "weights" of neural networks constitute a new form of data, and the demand for efficient storage and transmission of this modality is rapidly increasing. This demand is particularly pressing for large language models (LLMs), whose parameter counts range from hundreds of billions to the trillion scale (Gemini team, 2025). Several contexts highlight the challenge: intra- and inter-chip communication when serving foundation models (Pope et al., 2023); distributed training that relies on periodic exchange of weights and loss gradients (McMahan et al., 2017); and the storage of weight updates for models fine-tuned to enable personalization (Hu et al., 2022). As a result, developing effective compression techniques for language model weights has become a critical research direction to make massive, state-of-the-art models more practical and widely accessible.

Currently, the dominant paradigm in language model weight compression is the *quantization with minimal, handcrafted transforms*. In other words, most methods apply quantization either directly in the original weight space or in a transformed space defined by low-complexity, manually designed encoders and decoders. This trend is especially prevalent in the accelerating LLM inference on conventional GPUs, where the latency from complicated decoding operations can be a critical drawback. In this setting, the scalar quantization in the weight space with an advanced error compensation mechanism remains a strong baseline (Frantar et al., 2023), while quantization in per-channel-scaled or Hadamard-transformed spaces excels at lower bitrates (Lin et al., 2024; Ashkboos et al., 2024).

In this work, we take a step back from these restrictions, and ask:

> *"What advantages can nonlinear, learned transforms bring for compressing model weights?"*

There are several motivations for considering this question. First, modern deep learning is becoming increasingly memory-bound, dramatically lowering the relative cost of computation (Gholami et al., 2024). Second, it is a growing practice to utilize customized hardware, where decoding computations can be amortized using hard-encoded weights or in-memory processing (Mutlu, 2023).

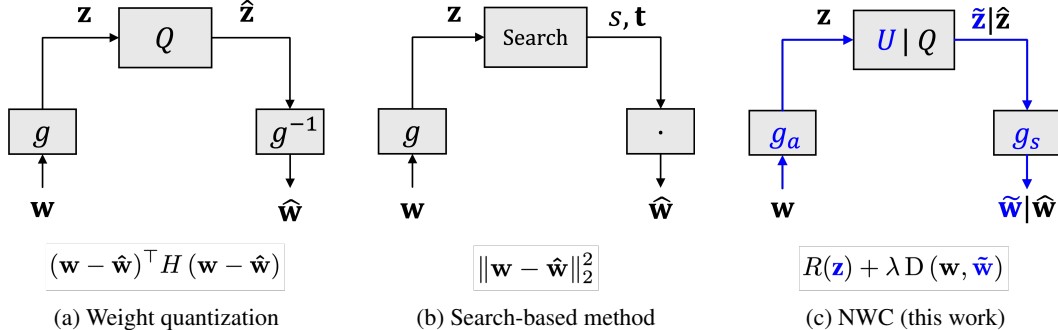

(a) Weight quantization    (b) Search-based method    (c) NWC (this work)

Figure 1: Operational diagrams of various weight compression paradigms, and corresponding minimization objectives. Learnable paths are marked in blue. $g, g^{-1}$ : denote the transforms and their inverse; $\cdot$: random matrix generation and multiplication; $U/Q$ : Uniform noise / Quantization; Search : grid search of codewords and coefficients; $g_a, g_s$ : trainable analysis & synthesis networks.

Third, handcrafting a linear transform requires much human labor through trial-and-error, making it difficult to extend for covering a diverse family of neural network weights with highly different distributional characteristics from each other (Pierro & Abreu, 2024; Xu & Yang, 2025).

**Contribution.** To this end, we develop a *neural compression* framework for the weights of language models. Specifically, we train neural networks to compress and reconstruct model weights using a dataset of pretrained weight tensors. This approach allows the neural codec to learn the distributional properties of language model weights and their impact on downstream performance—such as sensitivity to outlier activations (Sun et al., 2024a)—rather than relying on hard-coded handcrafted components, such as the Hadamard transform.

Simply put, our method, coined *neural weight compression* (NWC), adopts and adapts the encoder-decoder framework to the task of language model weight compression (Ballé et al., 2017). In doing so, two key obstacles emerge: Weight tensors highly vary in their sizes and scales, and using the downstream model performance as the training loss is difficult due to the size of typical language models. These challenges are addressed by NWC through several technical components, including:

- *Column-wise tensor chunking and normalization:* We preprocess the tensor weights by chunking them column-wise and normalizing them to uniform lengths and scale.
- *Importance-aware training loss:* We train the codec with randomly assigned saliency levels to each chunk of the model weight, in the form of the sample weight. At the inference phase, these scores are assigned based on the relative impact of the chunk on the model performance.
- *Inference-time error compensation:* At inference, we dynamically update the remaining weights to compensate for the errors introduced by the compression of previously-processed weights.

Empirically, the proposed NWC achieves state-of-the-art accuracy-compression tradeoffs at 4-6 bit-rates and competitive performance at sub-3 bits on the Llama family of models (Touvron et al., 2023; Grattafiori et al., 2024), under both data-free and calibration-based settings. Furthermore, these performance benefits are not confined to language models but also extend to vision encoders such as SigLIP (Zhai et al., 2023) and DINO (Oquab et al., 2023).

## 2 RELATED WORK

**Neural data compression.** Deep-learning-based approaches have been quite successful in the task of data compression (Yang et al., 2023). Unlike conventional handcrafted codecs, neural codecs do not require an accurate modeling of the data generation procedure, and can be trained to fit the training data directly instead. Furthermore, this optimization can be done using any loss function, and thus can maximize diverse quality measures of the data. These advantages have made neural compression particularly successful in compressing high-dimensional natural signals such as images (Ballé et al., 2017; He et al., 2022), videos (Lu et al., 2019; Jia et al., 2025), or audios (Zeghidour et al., 2021; Défossez et al., 2023). In particular, in image compression, neural codecs have achieved

state-of-the-art performance, not only in terms of the classical MSE loss (He et al., 2022; Liu et al., 2023a; Li et al., 2024), but also in terms of the perceptual quality of the generated reconstructions (Mentzer et al., 2020; Muckley et al., 2023). Inspired by this progress, our work aims to extend this successful paradigm of neural compression to the data domain of *language model weights*—the data with high dimensionality and with complicated generation procedure, whose quality should be measured by model performance, rather than MSE.

**Weight compression via neural codecs.** There have been several prior attempts to train neural codecs for compressing the language model weights. Banaei et al. (2023) have proposed an autoencoder-based framework to compress the weights of BERT, outperforming shallow matrix factorization approaches. Cheng et al. (2024) have trained variational autoencoders that can compress the weights of relatively small-scale networks, including an LSTM. More recently, Leconte et al. (2024) have proposed to compress language model weights using a residual VQ-VAE encoder. In the work, the compression is done by overfitting the codes to the weights in a similar manner to the implicit neural representation codecs (Chen et al., 2023), with careful tuning of the decoder weights for each tensor. These works are either applied to small-scale models only or adopt a piecemeal strategy, which requires a separate network to be trained for each individual weight matrix. In contrast, our work trains a single, unified neural network that holistically compresses the entire set of weights for an LLM-scale model. To the best of our knowledge, this represents the first demonstration of a global neural compression framework for large language models.

**Weight compression via quantization.** Quantization, *i.e.*, simply converting full-precision floating-point weights into low-bit integer representations, stands as a dominant technique for compressing pretrained language models. GPTQ (Frantar et al., 2023) leverages second-order information to iteratively quantize weights while updating the remaining ones to compensate for quantization errors. Building on this, AWQ (Lin et al., 2024) identifies and protects salient weights by incorporating activation-aware scaling. Another approach involves transforming weights to make them more amenable to quantization. Methods such as QuIP (Chee et al., 2023) and QuaRot (Ashkboos et al., 2024) utilize incoherent transformations, like the Hadamard transform, to reshape weight and activation distributions to be more Gaussian-like, thereby mitigating large outliers and improving quantization robustness. Pushing compression to its limits, vector quantization techniques have recently gained attraction. Methods like AQLM (Egiazarian et al., 2024), QuIP# (Tseng et al., 2024a), and QTIP (Tseng et al., 2024b) employ vector quantization and trellis coding to achieve extremely low bit-widths (e.g., sub-3-bit), capitalizing on the superior packing efficiency of these schemes. In contrast to these predominantly handcrafted methods, this work explores the potential of a learned approach that leverages data-driven knowledge, such as the heavy-tail distribution of weights. While recent works (Egiazarian et al., 2024; Cai et al., 2024; Tseng et al., 2024b) utilize learnable frameworks that update codebooks via gradient-based optimization, our framework fundamentally differs by leveraging nonlinear transformations and density estimation-based entropy coding.

**Other approaches.** There are many other non-learned approaches for compressing neural network weights. Pruning reduces the number of bits to represent the model by removing redundant parameters (Hassibi et al., 1993; Chee et al., 2022; Sun et al., 2024b). Another prominent direction is the low-rank matrix approximation, such as applying singular value decomposition to compress weight matrices in language models (Hsu et al., 2022; Yuan et al., 2023; Wang et al., 2025b; Lin et al., 2025). More recently, Shafipour et al. (2025) has explored representing weight blocks using a random seed, with corresponding coefficients as the compressed code. These works tend to have a very small computational footprint for decoding, using one or no matrix multiplications. In contrast, our work aims for a codec with an increased decoding complexity, which allows exploiting the computational capacities of modern hardware for a better compression-accuracy tradeoff.

## 3 BACKGROUND

### 3.1 PRELIMINARIES: VARIATIONAL DATA COMPRESSION

We first briefly describe the autoencoder-based compression framework of Ballé et al. (2017), which we closely follow in our design of the proposed neural weight compression algorithm (Figure 1c).

In a nutshell, the approach jointly trains two networks: an *analysis network* $g_a(\cdot)$ (a.k.a. encoder) which processes the input signal $\mathbf{x}$ into a continuous latent representation $\mathbf{z} = g_a(\mathbf{x})$, and a *synthesis*

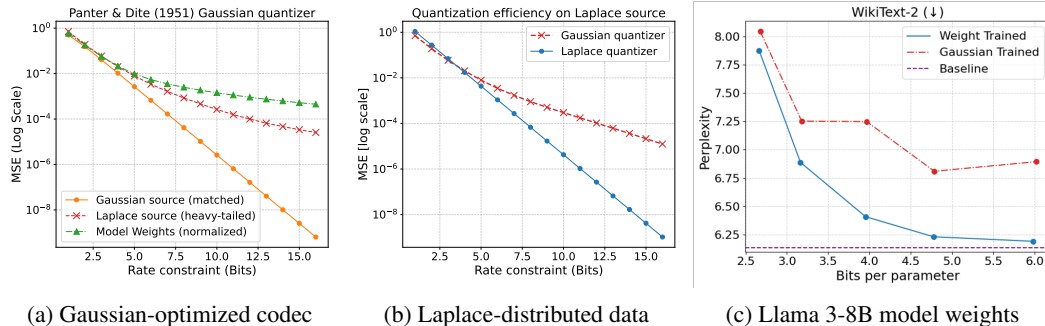

(a) Gaussian-optimized codec    (b) Laplace-distributed data    (c) Llama 3-8B model weights

Figure 2: Rate-distortion curves of codecs assuming heavy-tailed and Gaussian distributions. (a) When using a codec optimized for Gaussian data. (b) Comparing different codecs on Laplacian data. (c) Comparing of neural codecs trained on synthetic Gaussian data vs. on actual model weights.

*network* $g_s(\cdot)$ (a.k.a. decoder) which reconstruct the signal $\hat{\mathbf{x}} = g_s(\hat{\mathbf{z}})$ from a quantized, discrete latent $\hat{\mathbf{z}} = Q(\mathbf{z})$. Here, the *code*—i.e., a binary bitstream that represents the original signal in a compact manner—is generated by losslessly compressing the quantized latent $\hat{\mathbf{z}}$ via entropy coding, such as arithmetic or Huffman coding (Salomon, 2004).

The goal of data compression can be expressed as a constrained minimization problem: Minimize some distortion $\mathcal{L}_{\text{distortion}}$, given rate constraints. This is solved by optimizing its Lagrangian relaxation. Precisely, the encoder-decoder pair is jointly trained using the rate-distortion loss objective

$$\mathcal{L} = \mathcal{L}_{\text{rate}} + \lambda \cdot \mathcal{L}_{\text{distortion}}$$
$$= \mathbb{E}_{\mathbf{x} \sim \mathcal{D}}[-\log[p_{\hat{\mathbf{z}}}(Q(g_a(\mathbf{x}))] + \lambda \cdot d(\mathbf{x}, g_s(\hat{\mathbf{z}}))], \tag{1}$$

where $\lambda$ is a hyperparameter that balances two losses. Here, the rate loss $\mathcal{L}_{\text{rate}}$ measures the Shannon entropy of the quantized latents $\hat{\mathbf{z}}$ estimated with the (learnable) entropy model $p_{\hat{\mathbf{z}}}$, and the distortion loss $\mathcal{L}_{\text{distortion}}$ measures the distortion $d(\cdot, \cdot)$ between the original and reconstructed signal.

Due to the non-differentiability of the quantization function $Q(\cdot)$, the training is done by approximating the quantization procedure by adding a uniform noise $\mathcal{U}(-\frac{1}{2}, \frac{1}{2})$ to each dimension of the continuous latent $\mathbf{z}$, generating a noisy proxy $\tilde{\mathbf{z}}$.

**Advantages.** A notable strength of this framework is its flexibility. The approach allows the network to be optimized for any differentiable distortion measures, such as perceptual metrics (Zhang et al., 2018). Also, it admits a direct control over the rate-distortion trade-off by adjusting $\lambda$, alleviating the need for laborious searches for configurations to target bitrates. Furthermore, the framework is known to have relatively lower encoding and/or decoding cost when compared with other neural compression frameworks such as implicit neural representations (Kim et al., 2024).

### 3.2 Toy example: Tails hurt more at higher bitrates than lower

By learning directly from a dataset of pretrained weights, we expect our codec to be able to capture the distributional properties of the weights. The question is: exactly what advantage can we expect?

In this subsection, we provide a simple illustrative example suggesting that, for heavy-tailed distributions like neural network weights (Mahoney & Martin, 2019), the advantage may be the *better approximation at mid-to-high bitrates,* rather than at low bitrates. This observation is well-aligned with our experimental results in later sections, that the proposed NWC achieves particularly good performance at 4–6 bits. In a sense, this is in contrast with the strengths of typical advanced hand-crafted quantization techniques at very low bitrates (Ashkboos et al., 2024; Tseng et al., 2024b).

In Figure 2a, we plot the reconstruction error when we apply a codec optimized for Gaussian distribution to compress heavy-tailed data, such as Laplace distribution or the weights of the language model (here, we use the weights of Llama 3-8B). Specifically, we implement the asymptotically

optimal $b$-bits scalar quantizer for the Gaussian distribution by companding:

$$\text{Encoding:} \qquad x \mapsto k = \lfloor L\,\Phi(x) \rfloor, \tag{2}$$

$$\text{Decoding:} \qquad k \mapsto \hat{x} = \Phi^{-1}\big(\tfrac{k+0.5}{L}\big), \tag{3}$$

where $L = 2^b$. Here, $\Phi(x)$ is the companding function derived from the optimal point density $\Phi'(x) \propto p(x)^{1/3}$ (Panter & Dite, 1951), where $p(t)$ denotes the standard Gaussian PDF:

$$\Phi(x) = \frac{\int_{-\infty}^{x} p(t)^{1/3}\,dt}{\int_{-\infty}^{\infty} p(t)^{1/3}\,dt}, \tag{4}$$

From Figure 2a, we observe that the relative scale of the reconstruction error for Laplace distribution grows significantly higher than the Gaussian as we increase the bitrate. The trend is similar for Llama weights. As can be seen in Figure 2b, this is suboptimal compared to the Laplace-optimized codec, at the high-rate regime. From Figure 2a, we observe that the relative scale of the reconstruction error for Laplace distribution grows significantly higher than the Gaussian as we increase the bitrate. The trend is similar for Llama weights. As can be seen in Figure 2b, this is suboptimal compared to the Laplace-optimized codec, at the high-rate regime.

Crucially, this trend in reconstruction error translates to downstream model performance, as we observe in Figure 2c. Here, we compare the compression performance of the same neural weight compression model when trained separately on synthetic Gaussian data, versus actual model weights.

Such an advantage can be explained as follows: At very low bitrates, the fact that the Gaussian-fitted codec does not assign many centroids at the tail of the distribution is not too problematic, as the bitwidth is barely enough to cover the central area of the distribution. However, as the bitwidth increases, the drawback from not assigning enough centroids to the tails becomes more visible.

## 4 NEURAL WEIGHT COMPRESSION

In this section, we describe the proposed neural weight compression (NWC) framework for compressing the language model weights. In a nutshell, the NWC consists of three main steps.

- **Preprocessing** (Section 4.1). For both training and inference, we process weight tensors into vectors of regular sizes and scales, allowing the model to handle tensors with diverse characteristics.
- **Training** (Section 4.2). We train the codec using a loss that can account for various importance levels of weight chunks, applying importance augmentations for a better sample efficiency.
- **Inference** (Section 4.3). NWC compensates for the errors incurred by compressing weights by updating the weights that are not compressed yet, via both Hessian-based update and fine-tuning.

### 4.1 PREPROCESSING: CHUNK AND NORMALIZE

Unlike images or videos, weight tensors in language models can vary highly in their structural properties, such as their dimensionalities and scales. For example, for the Llama 3-8B model, the key projection layer has a size $\mathbb{R}^{1024 \times 4096}$, while the up projection layer has a size $\mathbb{R}^{14336 \times 4096}$.

To handle such heterogeneity, we preprocess each weight tensor in three steps. First, we partition each weight tensor into column vectors. Then, we normalize each column vector to have a standard deviation of 1 by multiplying the reciprocal of the standard deviation for each column vector. Finally, we chunk the column vectors into vectors of uniform length $d = 16$ (Figure 3, left).

$$\mathbf{W} \in \mathbb{R}^{m \times n} \quad \xrightarrow{\text{partition}} \quad \mathbf{w}_{\text{col}} \in \mathbb{R}^{m} \quad \xrightarrow{\text{normalize}} \quad \bar{\mathbf{w}}_{\text{col}} \in \mathbb{R}^{m} \quad \xrightarrow{\text{chunk}} \quad \mathbf{w} \in \mathbb{R}^{16} \tag{5}$$

The normalization factor for each column is stored in FP16 and used to restore the scale during reconstruction, incurring a negligible overhead of roughly 0.004 bits per parameter.

Here, we select column-wise chunking—instead of row-wise—due to the ease at the inference stage. The importance score, which will be used to assign different desired importance levels for each value, will be computed for each column by design. By decomposing into columns, we use a

single importance score for the whole chunk. Also, the relatively small chunk size helps reduce the memory needed for the error compensation procedure at the inference phase. See Section 4.3 for more details.

With these preprocessing steps, the proposed NWC can handle the weight tensors of various language models with a single pair of compact encoder and decoder networks.

## 4.2 TRAINING: MINIMIZE THE IMPORTANCE-AWARE TRAINING LOSS

The proposed NWC aims to minimize the error of the model that uses the compressed weights, conditioned on the rate constraint. More concretely, we consider $\mathcal{L}_{\text{model}}(\hat{\mathbf{w}}) = \text{error}(\hat{\mathbf{w}}, \mathcal{T})$, where $\hat{\mathbf{w}}$ denotes the compressed weights and $\text{error}(\cdot, \mathcal{T})$ denotes the error of the model parametrized by weights on some downstream task $\mathcal{T}$. This is in contrast with the usual data compression literature, where it is popular to use the mean-squared error as the distortion.

However, using $\mathcal{L}_{\text{model}}(\hat{\mathbf{w}})$ directly for training is challenging, as computing the gradient with respect to the loss needs evaluating second derivatives with LLM-scale parameters, which require huge computation and memory. Thus, we use the following approximate minimization objective: Pick a linear layer $\mathbf{x} \mapsto \mathbf{W}\mathbf{x}$, and consider the distortion in the layer, which can be approximated as

$$\mathbb{E}[\|\mathbf{W}\mathbf{x} - \hat{\mathbf{W}}\mathbf{x}\|_2^2] \approx \text{tr}((\mathbf{W} - \hat{\mathbf{W}})\mathbf{H}(\mathbf{W} - \hat{\mathbf{W}})^\top)$$

$$\approx \text{tr}((\mathbf{W} - \hat{\mathbf{W}})\text{diag}(\mathbf{H})(\mathbf{W} - \hat{\mathbf{W}})^\top) =: d(\mathbf{W}, \hat{\mathbf{W}}), \quad (6)$$

where $\mathbf{H}$ denotes the loss Hessian, which can be computed from $m$ input activations $\{\mathbf{x}_i\}_{i=1}^m$ via $H = \frac{1}{m}\sum_{i=1}^m \mathbf{x}_i\mathbf{x}_i^\top$; see, e.g., Frantar et al. (2023). The second approximation is essentially due to LeCun et al. (1989), and recent observations suggest that the approximation tends to be accurate in the context of language models (An et al., 2025; Sun et al., 2024a). This approximation essentially boils down to the input activation scaling of Sun et al. (2024b).

Motivated by the fact that $d(\mathbf{W}, \hat{\mathbf{W}})$ is simply a weighted version of the squared error, the proposed NWC trains by minimizing the ***importance-aware loss***:

$$l_{\text{Imp}} = l_{\text{rate}} + \lambda_I^{(\mathbf{w})}\lambda \cdot \text{MSE}(\mathbf{w}, \hat{\mathbf{w}}). \quad (7)$$

Here, $\lambda_I^{(\mathbf{w})}$ is a variable *importance level*, which represents the relative importance of each weight chunk. Precisely, we select the importance level among a set of $K$ discrete levels $\{\lambda_1, \ldots, \lambda_K\}$. Empirically, it turns out that a small $K$ is sufficient for all LLMs tested (e.g., 2 or 4).

During the training, the importance level for each chunk is drawn uniformly at random, which works as an effective data augmentation technique. The index for the drawn importance level will be given to the encoder as an input, so that the encoder may learn to infer how much information should be compressed away for the given chunk, from the given index.

During the inference, we select the importance level depending on the corresponding Hessian diagonal computed from calibration samples. We give more details in Section 4.3.

**Network architectures.** We provide a visual overview of the proposed NWC model architectures in the right panel of Figure 3. Both the analysis network $g_a$ and the synthesis network $g_s$ consist of multiple residual MLP blocks. To condition the compression process on the desired importance level, the discrete index is first converted into a dense vector via a learnable embedding. This embedding is then element-wise multiplied with the hidden state after each residual block in both the encoder and the decoder. This allows the networks to learn variable-quality transformations within a single model. The importance levels, stored as a single integer, $\lceil \log_2(l) \rceil$ bits per column of the weight matrix, result in a negligible rate overhead of less than 0.001 bits per weight. For the entropy model, we employ non-parametric, fully factorized density model proposed by Ballé et al. (2017), coupled with arithmetic coding.

## 4.3 INFERENCE: COMPENSATE FOR COMPRESSION ERROR

At inference time, we compress each weight tensor by processing the weights with corresponding importance level. To determine the importance level, we first compute the Hessian diagonal for each weight matrix. Then, based on which quantile of the Hessian diagonal of each column vector

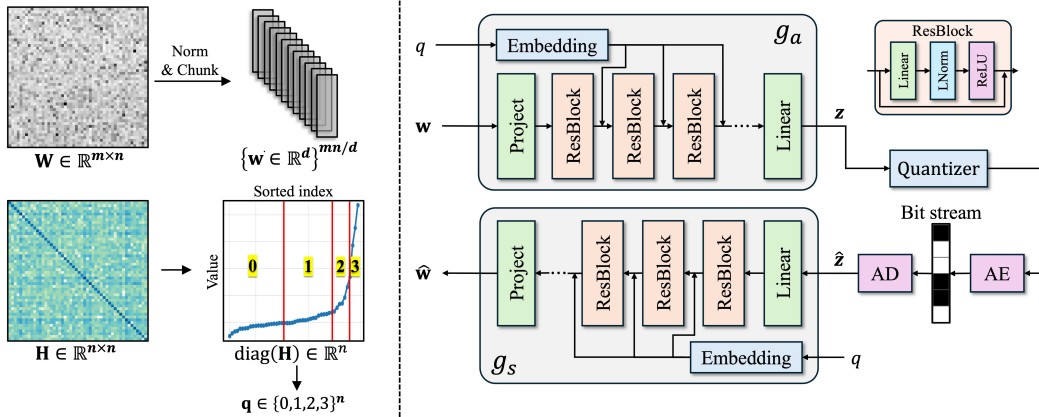

Figure 3: A visual description of the proposed neural weight compression (NWC) framework. (Left) Preprocessing steps for the weight tensors, including column-wise chunking and normalization, and importance level assignment. (Right) Model architectures of the analysis and synthesis networks. AE/AD refer to Arithmetic Encoding/Decoding.

belongs to, we assign the importance level; we use the logarithmic scale, as illustrated in the left panel of Figure 3.

Furthermore, we adopt two additional mechanisms to improve the quality of the compressed model by compensating for the compression error by updating the weights that are not compressed yet.

**Intra-layer error feedback.** While compressing a weight tensor, we update the column vectors that are not compressed yet to minimize the distortion in the layer output. In particular, we incorporate the LDLQ objective proposed by Chee et al. (2023), to update the $k$-th column weight vector $\mathbf{w}_k$ as

$$\tilde{\mathbf{w}}_k = \mathbf{w}_k + (\mathbf{W}_{1:k-1} - \hat{\mathbf{W}}_{1:k-1})\mathbf{a}_k, \tag{8}$$

where $\mathbf{W}_{1:k-1}$ denotes the first $k-1$ columns of the weight tensor, $\hat{\mathbf{W}}_{1:k-1}$ denotes its compressed version, and $\mathbf{a}_k$ denotes the $k$-th column of the $\mathbf{L}^\top - \mathbf{I}$; here, $\mathbf{L}$ is the upper trianglular matrix from the LDL decomposition of the loss Hessian for this layer. The updated $\tilde{\mathbf{w}}_k$ will then be compressed by the neural codec, and the next column will be updated subsequently. We note that our column-wise chunking strategy makes the compression pipeline compatible with this procedure.

**Inter-layer recovery fine-tuning.** Before compressing each layer in a transformer block, we fine-tune the layer to account for the compression conducted on other layers in the same block; this is inspired by recent works in model compression (Tseng et al., 2024a; Ding et al., 2025; Malinovskii et al., 2024). Precisely, after compressing each layer in a $k$-th block, we fine-tune the uncompressed layers in the same block to match the block output to the $k$-th block output of the uncompressed model. Here, as the block input, we use the features of the uncompressed model computed from calibration data; this enables a parallel compression of multiple transformer blocks. This block-wise method can be complemented by a final end-to-end fine-tuning of the entire model after compression is complete, similar to the strategy in QuIP#.

## 5 EXPERIMENTS

### 5.1 EXPERIMENTAL SETUP

**Baselines.** We compare NWC against post-training model compression methods, including (1) Scalar weight quantization methods: AWQ (Lin et al., 2024), GPTQ (Frantar et al., 2023), and SpinQuant (Liu et al., 2025); (2) Vector quantization methods: QuIP# (Tseng et al., 2024a) and QTIP (Tseng et al., 2024b); (3) Pseudo-random generator method: SeedLM (Shafipour et al., 2025); (4) Neural codec: ReALLM (Leconte et al., 2024). Other baselines can be found in Appendix B.3

For a thorough comparison, we reproduce the results of the baselines over the bitwidths missing in the original papers; see Appendix A.1 for details. Furthermore, to evaluate the efficacy of the

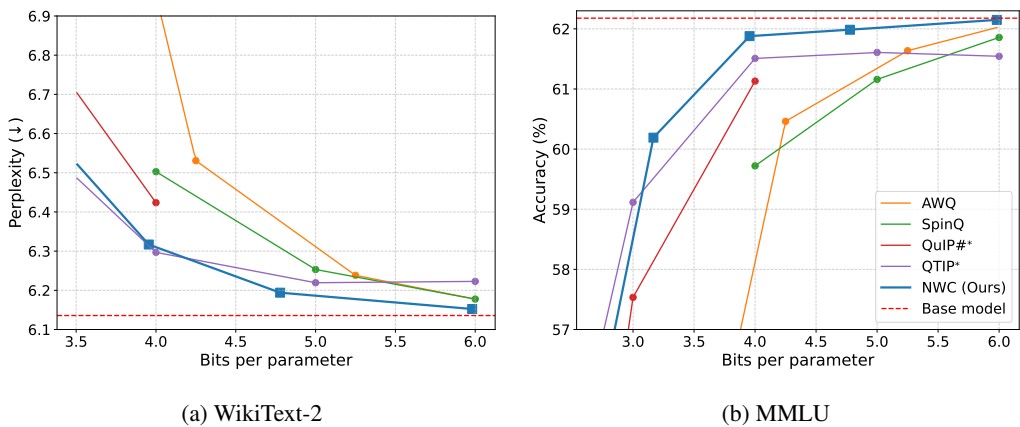

(a) WikiText-2                    (b) MMLU

Figure 4: Compression results of Llama 3-8B across various bitrates. (a) WikiText-2 perplexity results with context length of 2048. (b) Zero-shot accuracy of MMLU benchmark.

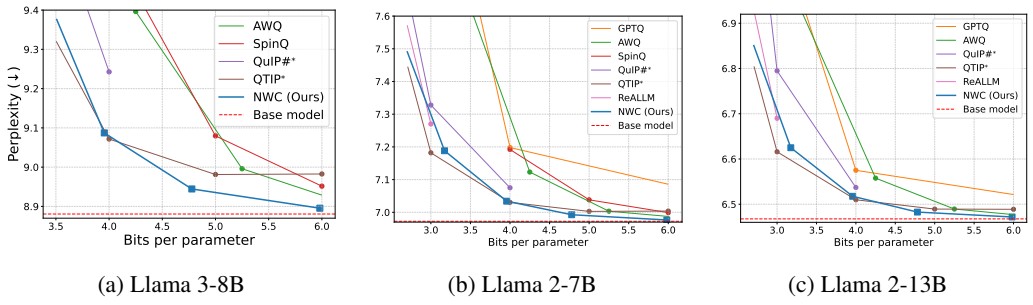

(a) Llama 3-8B                (b) Llama 2-7B                (c) Llama 2-13B

Figure 5: C4 perplexities results of Llama 2 and Llama 3 models

compression scheme in isolation, we compare with the baselines without end-to-end fine-tuning. To clarify this point, we mark the modified baselines—QuIP# and QTIP—with an asterisk (*).

**Evaluation.** To evaluate the quality of compressed models, we measure both perplexities and zero-shot accuracies. The perplexity is measured on the WikiText-2 (Merity et al., 2016) and the C4 datasets (Raffel et al., 2020) with the context length of 2048. Zero-shot accuracies are measured across both MMLU benchmark (Hendrycks et al., 2021) and 6 common-sense reasoning tasks (ARC-Easy, ARC-Challenge, WinoGrande, PiQA, HellaSwag, BoolQ).

**Training.** We train compression network on a dataset consisting of all linear layer weight tensors from Llama 3-8B. Both the encoder and decoder of the model consist of 4-layer residual MLPs with a width of 512. We have used four quality levels: $\lambda_q \in \{0.29, 0.83, 10, 20\}$. To plot the rate-distortion curve, we use $\lambda$ ranging from 30 to 10,000. See Appendix A.3 for more details.

### 5.2 LLM WEIGHT COMPRESSION

In Figure 4, we measures the quality for compressed Llama 3-8B models (Grattafiori et al., 2024) at various bit-rates. NWC consistently outperforms most quantization-based approaches, achieving lower perplexity and higher accuracy. When compared with QTIP, the advantage of the neural approach becomes more pronounced at higher bit-rates, aligning with the Section 3.2.

We further demonstrate the generalizability of our approach across different architectures, including Llama 2 (Figure 5) and MoE-based models (Figure 6). Without retraining the codec, the performance on these models is similarly strong in the 4-6 bit regime to that observed on Llama 3-8B.

### 5.3 DATA-FREE SCENARIOS

Additionally, we evaluated the performance of NWC in a data-free scenario, where the model is compressed without any calibration data. For this experiment, we compared our method against

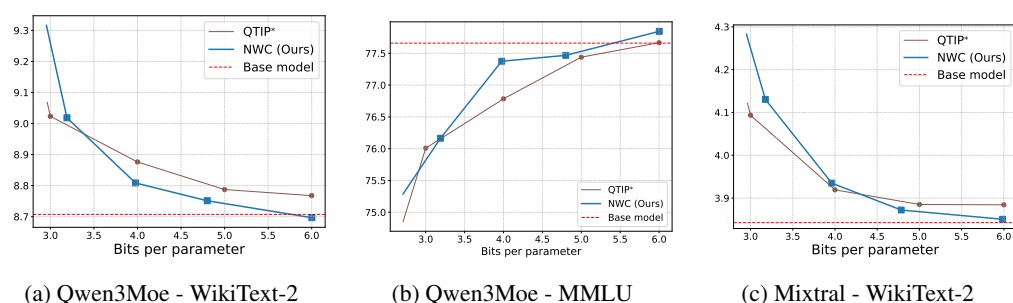

(a) Qwen3Moe - WikiText-2   (b) Qwen3Moe - MMLU   (c) Mixtral - WikiText-2

Figure 6: Perplexity and zero-shot accuracy results of Qwen3-30B-A3B and Mixtral-8x7B-v0.1.

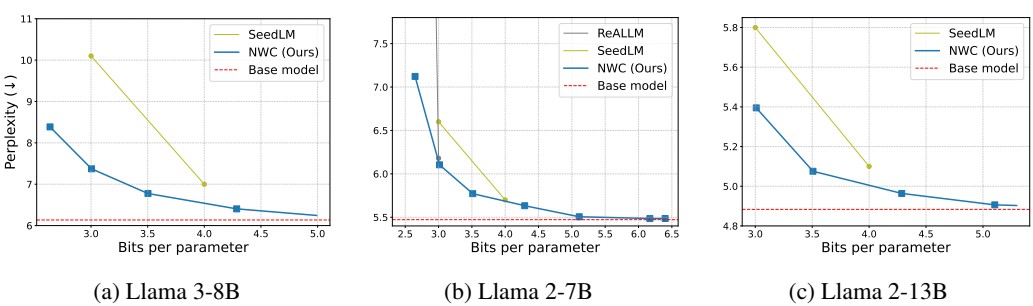

(a) Llama 3-8B   (b) Llama 2-7B   (c) Llama 2-13B

Figure 7: WikiText-2 perplexity results of Llama models without calibration data

SeedLM (Shafipour et al., 2025) and data-free version of ReALLM in the paper (Leconte et al., 2024). We used the results as reported in their original paper. To ensure a fair comparison, we used a simplified version of NWC where all weight blocks were compressed to a uniform quality level, without applying adaptive error feedback or block-wise fine-tuning.

As shown in Figure 7, NWC consistently achieves lower perplexity compared to SeedLM across all models. This suggests that a learnable approach can form a more effective representation of the weight data than a simple approximation based on a heuristic search. As reported in the paper, ReALLM exhibits a significant performance drop at low bit-rates in the data-free setting, as it heavily relies on the tuning of low-rank components using data.

## 5.4 BEYOND LANGUAGE MODEL

To demonstrate the versatility of NWC, we evaluate our methods on the prominent vision encoders. Specifically, we evaluate zero-shot classification accuracy of CLIP-ViT-L/16 (Radford et al., 2021) and SigLIP-B/16 (Zhai et al., 2023), and linear probe accuracy of DINOv2-L (Oquab et al., 2023) on ImageNet (Russakovsky et al., 2015). For these experiments, the NWC codec is first pre-trained on Llama weights and then fine-tuned on the weights of each respective model. The per-layer Hessian is calculated using samples from the Conceptual Captions dataset (Sharma et al., 2018). No recovery fine-tuning is applied to either method, which we mark as ($^\dagger$). See Appendix A.4 for more details.

The results, presented in Figure 8, show that NWC achieves superior performance at mid-to-high bit-rates. This is consistent with the trend observed in the LLM experiments, suggesting that the benefits of our neural approach generalize across different model architectures and modalities.

## 5.5 ABLATION STUDY

As shown in Figure 9, we evaluate the contribution of each proposed component. We observe that finer-grained (i.e., channel-wise) normalization offers the most favorable rate-distortion trade-off (Figure 9a), and that leveraging importance-awareness is crucial for minimizing perplexity at low bit-rates (Figure 9b). Additionally, the results confirm that both intra- and inter-layer error compensation mechanisms are essential for recovering model performance after compression (Figure 9c).

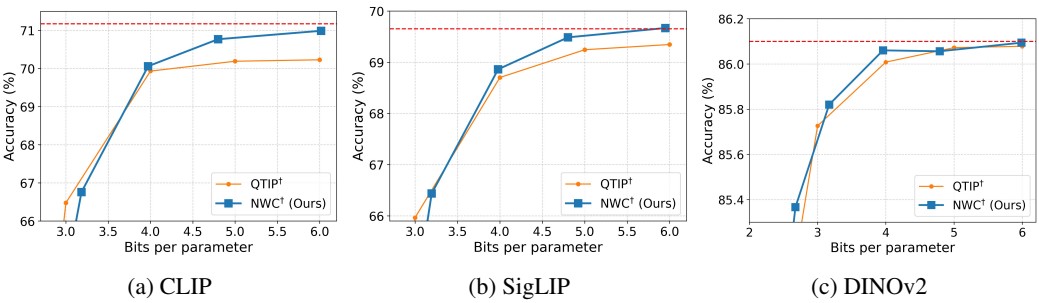

(a) CLIP        (b) SigLIP        (c) DINOv2

Figure 8: ImageNet-1k classification accuracy of vision models. We report zero-shot classification with text prompts for CLIP and SigLIP, and linear probing accuracy for DINOv2.

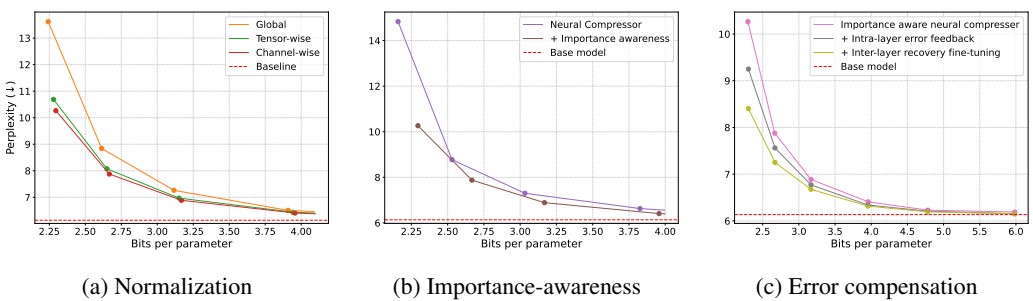

(a) Normalization        (b) Importance-awareness        (c) Error compensation

Figure 9: WikiText-2 perplexity results of LLaMA models with context length of 2048.

## 5.6 OTHER EXPERIMENTS

In Appendix B, we provide additional experimental results including evaluations on common-sense reasoning tasks (Figure 10), a comparison over a wider bit-rate range (Figure 11), ablation studies for encoder/decoding network and entropy model (Appendix C), the encoding and decoding latency (Table 3), and an analysis of the function of learned non-linear transformation (Appendix D.3)

## 6 CONCLUSION

In this work, we introduce Neural Weight Compression (NWC), a novel framework that represents one of the pioneering steps in applying the neural compression paradigm directly to the weights of large-scale models. This framework targets broader bottlenecks in efficient storage and transmission. Regarding in-GPU inference, our approach is particularly advantageous for large-batch serving scenarios, where decompression overhead is effectively amortized across concurrent requests.

**Limitation and future work**. A key limitation of this work is that the utility of NWC has been demonstrated primarily through the lens of the rate-accuracy tradeoff, as practical inference-time acceleration is currently infeasible without dedicated system or hardware support for the neural decoder. Co-designing hardware-accelerated solutions is a crucial direction for future work.

While our results show that NWC generalizes well across several language models, the scope is limited when considering the wide variety of modern model architectures. Investigating the generalization and adaptation of neural weight compression to networks with different characteristics (Pierro & Abreu, 2024; Xu & Yang, 2025) will be an important future direction, which is essential to assess the broader potential and foster the growth of the neural weight compression field.

NWC framework opens up the possibility of leveraging a wide range of recent advances from the broader neural data compression literature. For instance, incorporating techniques such as progressive coding (Jeon et al., 2023; Lu et al., 2021; Lee et al., 2022) and direct code optimization (Gao et al., 2022; Perugachi-Diaz et al., 2024) are promising directions for future work. Exploring these avenues could further enhance both the performance and flexibility of the NWC framework.

## REPRODUCIBILITY STATEMENT

We reproduced the results for all baselines in our experiments using their official code repositories. Following standard practice in the compression literature, the LLMs were sourced from publicly available checkpoints, such as those on HuggingFace, and the datasets for model evaluation were identical to those used in the benchmark studies. To ensure the reproducibility of our own methodology, we plan to release our code upon the publication of the final version of this paper.

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

# APPENDIX

## A    IMPLEMENTATION DETAILS

### A.1    DETAILS ON REPRODUCTION OF BASELINES OVER VARIOUS BITWIDTHS

For QTIP, AWQ, and SpinQuant, we reproduce the results of the baselines over the bitwidths missing in original papers.

- QTIP: We change $k$ in hybrid-computed codes
- SpinQuant: We optimize the W$b$A16KV16 rotation scheme for each target bitwidth $b$, then apply PTQ using this rotation in $b$-bits weight quantization setting.

- AWQ: We apply $b$-bit weight quantization using group size of 128, which induces roughly 0.25 extra bits per parameter.

In contrast, we were unable to reproduce results for QuIP# at arbitrary bit-widths. This is because it relies on fixed lattice codebooks for each specific bit-rate targeted in the original paper, which does not allow for flexible bitrate adjustment.

## A.2 HESSIAN GENERATION

In each experiment, the same Hessian matrices are used for NWC, QuIP#, and QTIP. The Hessian matrices were computed following QuIP# (Tseng et al., 2024a). For both Llama 2 and Llama 3 models, the Hessian was estimated using 6,144 sequences of length 2,048, sampled from the RedPajama dataset (Weber et al., 2024).

## A.3 NETWORK DESIGN AND TRAINING

**Hyperparmeter**. Details regarding the architecture and training of our compression network are provided in Table 1. We employ an auxiliary loss to update the quantile parameters of the entropy model, which is trained separately from the main rate-distortion objective.

Table 1: Hyperparameters for Network Design and Training

| Hyperparameter | Value |
| --- | --- |
| Block size | 16 |
| $g_a$ network width | 512 |
| $g_a$ number of residual blocks | 4 |
| $g_s$ network width | 512 |
| $g_s$ number of residual blocks | 4 |
| Entropy model channel size | 16 |
| Learning rate | $1 \times 10^{-4}$ |
| Learning rate (auxiliary loss) | $1 \times 10^{-3}$ |
| Optimizer | Adam |
| $\lambda$ | $\{30, 50, 100, 300, 1000, 10000\}$ |

**Pre-training.** We train the codec with the following compute resources and data:

- Compute cost: The codec is trained for 60 epochs, requiring 11.45 hours on a single NVIDIA A6000 Ada GPU.
- Dataset: The codec training does not require any external datasets or text corpora, utilizing only the model weights. For computational efficiency during random sampling, we aggregated 64 weight chunks into a single training sample. Consequently, for LLaMA-3-8B, the training set consists of approximately 6.8 million examples, with a validation set of 1,000 samples.

**Fine-tuning.** We fine-tune the codec on the target model weights for approximately 1 hour on a single NVIDIA A6000 Ada GPU.

## A.4 EXPERIMENT DETAILS FOR VISION ENCODER

We applied our compression technique to the vision encoder models as follows:

- CLIP and SigLIP, compression was applied to all projection layers within both the text encoder and vision encoder blocks.
- DINOv2, compression was applied to all layers within the main encoder blocks.
- For all models, the collection of weights targeted for compression was used as a dataset to fine-tune the NWC codec specifically for that model.

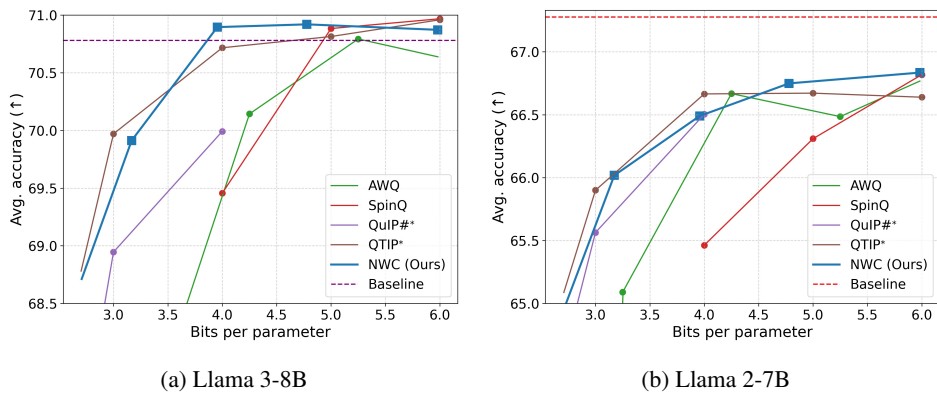

(a) Llama 3-8B

(b) Llama 2-7B

Figure 10: Average zero-shot accuracy of Llama 3-8B, Llama 2-7B on commonsense tasks ((ARC-Easy, ARC-Challenge, WinoGrande, PiQA, HellaSwag, BoolQ))

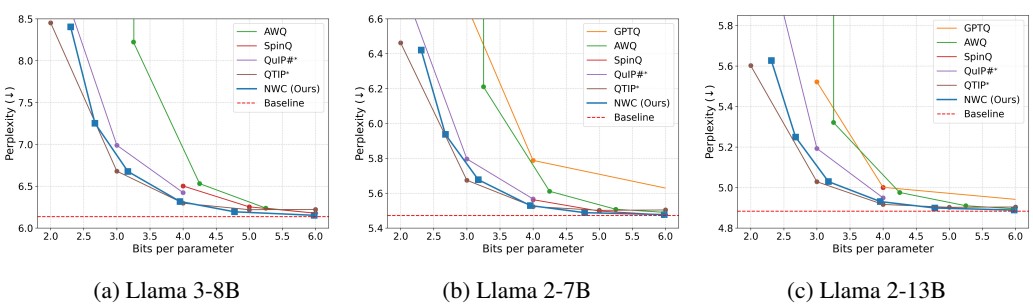

(a) Llama 3-8B

(b) Llama 2-7B

(c) Llama 2-13B

Figure 11: WikiText-2 perplexity results of LLaMA models with context length of 2048.

# B ADDITIONAL RESULTS

## B.1 ZERO-SHOT ACCURACY ON COMMON SENSE TASKS

Figure 10 shows the average zero-shot accuracy across various bit-rates. The accuracies are measured on a suite of six common-sense reasoning tasks: ARC-Easy, ARC-Challenge (Clark et al., 2018), WinoGrande (Sakaguchi et al., 2020), PiQA (Bisk et al., 2020), HellaSwag (Zellers et al., 2019), and BoolQ (Clark et al., 2019). We use the LM evaluation harness (Gao et al., 2021) of version 0.4.4.

Across the evaluated bit-rates, NWC demonstrates competitive performance against state-of-the-art quantization methods.

## B.2 FULL BITRATE RANGE RESULTS

Figure 11 presents the WikiText-2 perplexity results across a wider bit range. The results show that NWC outperforms most of the competing compression techniques. However, in the extreme-low bit regime (below 3 bits), QTIP exhibits superior performance.

## B.3 ADDITIONAL COMPARISON WITH NON-PTQ METHODS

In the main experiments, we leave out comparing NWC with the QAT methods The main reason is that QAT methods often require significantly more computation than ours. Specifically, LLM-QAT demands approximately 280 hours on an A100 80G. Furthermore, QAT relies on the full pre-training corpus or extensive synthetic data, NWC requires only the small calibration set.

**QAT & Neural codec.** In this section, we compare NWC against Quantization-Aware Training: LLM-QAT (Liu et al., 2023b), BitDistiller (Du et al., 2024) and other neural codecs: Re-

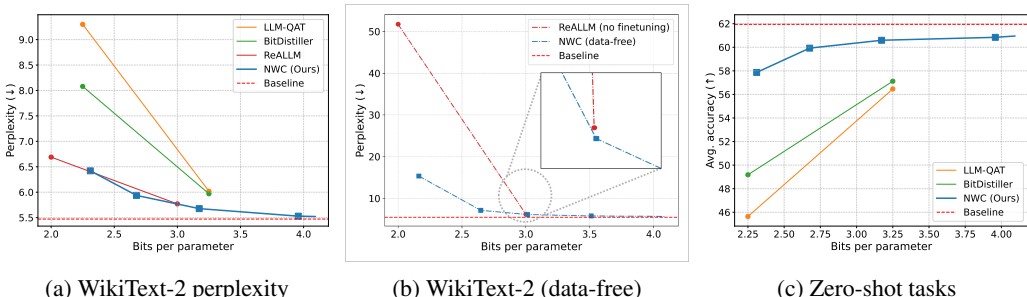

(a) WikiText-2 perplexity       (b) WikiText-2 (data-free)       (c) Zero-shot tasks

Figure 12: Compression results of Llama 2-7B.

Table 2: Performance of Llama 3-8B compressed by SVD-based methods and NWC.

| METHOD | Ratio | Wiki-2↓ | OpenQ | ARCe | WG | HS | PQ | MathQ | Avg↑ |
|--------|-------|---------|-------|------|-----|-----|-----|-------|------|
| Original | 0% | 6.14 | 0.35 | 0.80 | 0.73 | 0.60 | 0.80 | 0.40 | 0.61 |
| SVD-LLM | 20% | 11.82 | 0.29 | 0.77 | 0.64 | 0.51 | 0.72 | 0.30 | 0.54 |
| SVD-LLM V2 | 20% | 8.01 | 0.33 | 0.79 | 0.70 | 0.58 | 0.77 | 0.36 | 0.59 |
| NWC | 75% | **6.32** | **0.34** | **0.80** | **0.75** | **0.60** | **0.79** | **0.39** | **0.61** |

ALLM (Leconte et al., 2024). The evaluation is performed on Llama2-7B, measuring perplexity on WikiText-2 and the average zero-shot accuracy across four common-sense tasks (PiQA (Bisk et al., 2020), HellaSwag (Zellers et al., 2019), WinoGrande (Sakaguchi et al., 2020), and ARC-Challenge (Clark et al., 2018)). As shown in Figure 12, NWC consistently demonstrates superior performance compared to QAT-based methods at equivalent bit-rates. In contrast, ReALLM holds an advantage in the 2-bit regime. This is attributable to its reliance on low-rank component tuning. Crucially, when evaluated in a calibration data-free setting, NWC compression performance is substantially better.

**SVD-based methods.** Table 2 presents a comparison between NWC and SVD-based compression methods Wang et al. (2025b;a). We evaluate both the compression ratio and the zero-shot accuracy on a range of tasks, including OpenbookQA (Mihaylov et al., 2018) and MathQA (Amini et al., 2019). The results indicate that NWC achieves superior performance compared to SVD even at higher compression ratios.

## B.4 ENCODING & DECODING LATENCY ANALYSIS

We evaluate the computational efficiency of NWC in comparison to baseline methods. Table 3 details the wall-clock time of the encoding and decoding. The reported values represent the average wall-clock time ($\pm$ standard deviation) required to process a weight tensor of size $4096 \times 4096$. The entropy decoding latency of NWC is benchmarked using ANS codec of GPU-based data compression library[1].

Table 3: Encoding and decoding latency for $4096 \times 4096$ tensor. Measurements were conducted on a using an NVIDIA RTX 6000.

| Operation | NWC (Ours) | GPTQ | QuIP# | QTIP |
|-----------|-----------|------|-------|------|
| Encoding (sec) | 1.64 $\pm 0.30$ | 0.69 $\pm 0.02$ | 21.71 $\pm 0.82$ | 19.84 $\pm 0.30$ |
| Decoding (ms) | 3.88 $\pm 0.58$ | 0.08 $\pm 0.04$ | 2.94 $\pm 0.17$ | 2.32 $\pm 0.16$ |
|    Entropy Decoding (ms) | 2.37 | - | - | - |
|    Synthesis ($g_s$) | 1.51 | - | - | - |

---

[1] https://developer.nvidia.com/nvcomp

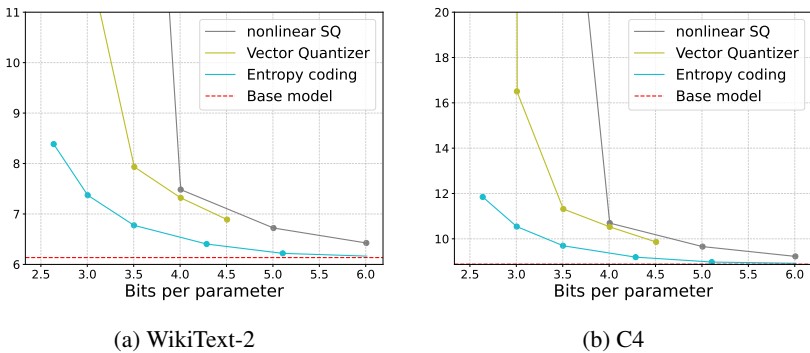

(a) WikiText-2                                 (b) C4

Figure 13: Ablation of entropy coding. Perplexity of Llama 3-8B on (a) Wikitext-2 (b) C4 dataset.

## C    ABLATION STUDIES

### C.1    EN/DECODER NETWORKS

To quantify the benefit of the learnable $g_a$ and $g_s$ networks, we compare the model against an entropy-only baseline. This baseline replaces $g_a$ and $g_s$ with identity functions and is trained using only the auxiliary and rate losses; without a distortion loss, it is incapable of controlling the bitrate or adapting reconstruction quality. For this ablation, both importance awareness and error compensation were disabled.

The results in Table 4 demonstrate that the learnable networks offer a large reduction in perplexity compared to the entropy-only model.

Table 4: Wikitext-2 and C4 perplexity for Llama-3-8B.

| En/Decoder | Bits | Wikitext-2↓ | C4↓ |
|---|---|---|---|
| Identity$(\cdot)$ | 2.12 | 27.8 | 33.2 |
| $g_a/g_s$ | 2.13 | 14.5 | 20.7 |

### C.2    USE OF ENTROPY ENCODING

To validate the effectiveness of density estimation-based entropy coding within our framework, we conducted an ablation study comparing our method against baselines where the entropy model is replaced by non-linear fixed-length quantizers. We trained and compared compression models utilizing a non-linear scalar quantizer, a vector quantizer, and the entropy model, respectively. For all three models, the encoder and decoder architectures were identical, and no input importance was employed. The two fixed-length quantizers were trained using the Straight-Through Estimator (STE), with the dimension of the vector quantizer set to 2.

As shown in Figure 13, utilizing entropy coding demonstrates significantly better performance compared to the other two models.

## D    ADDITIONAL ANALYSES

### D.1    PER-LAYER STATISTIC OF LARGE LAUGUAGE MODEL

In Figure 14, we visualize the kurtosis and standard deviation across different layer depths and types. From these results, we observe three key characteristics:

• All layers exhibit kurtosis values higher than that of a Gaussian distribution (i.e., $> 0$), indicating heavy-tailed distribution.

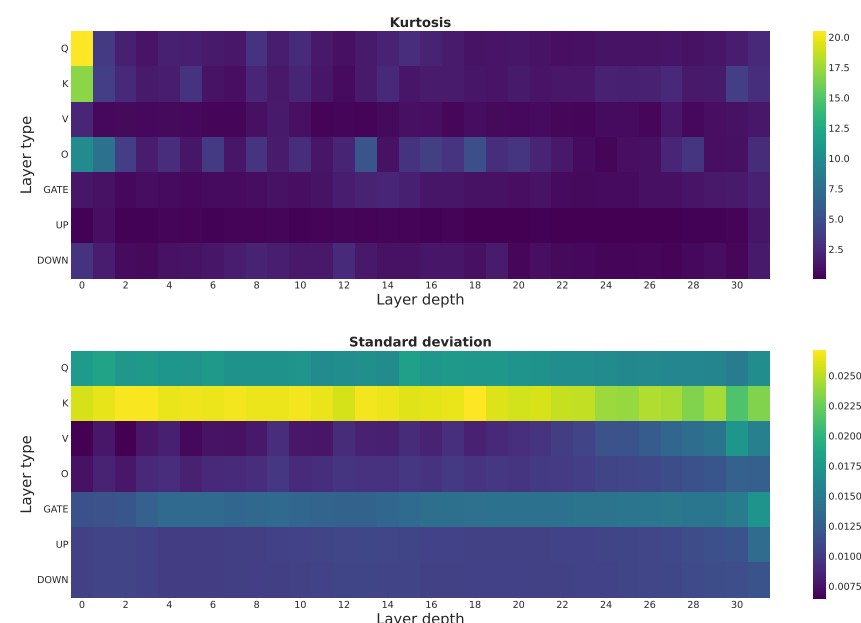

Figure 14: Layer-wise statistic of Llama 3-8B

- Certain layers exhibit extremely high kurtosis. This is particularly pronounced in the $Q$ and $K$ projections of the first block.
- The standard deviation varies depending on the layer type.

### D.2 NWC IS BETTER AT HEAVY-TAILED DISTRIBUTION

Figure 15 presents the rate-distortion curves of QTIP and NWC for a single layer. Our method outperforms QTIP in terms of MSE, particularly in layers characterized by high kurtosis (i.e., heavy-tailed distributions). This demonstrates that learned codecs surpass handcrafted approaches on heavy-tailed distributions, aligning with the observations in Section 3.2.

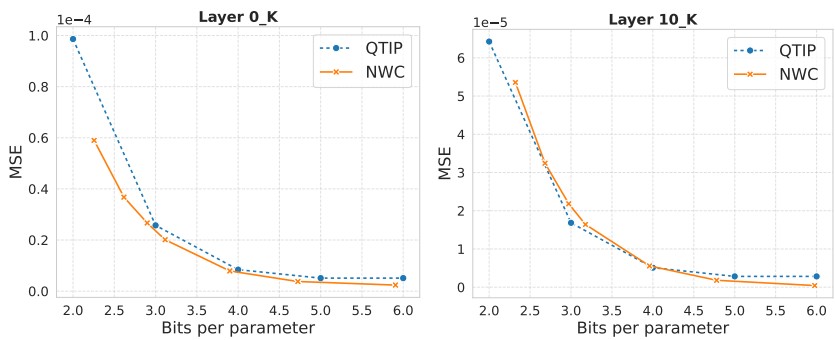

Figure 15: Per-layer rate-distortion curve of K projection layer in different blocks.

### D.3 LEARNED ENCODER EFFECTIVELY SUPPRESSES OUTLIERS

In Table 5, we compare the statistical properties of weights transformed by various methods, including Discrete Cosine Transform (DCT), Randomized Rotation, Randomized Hadamard Transform (RHT), and the learned neural encoder. The results indicate that the neural encoder is the most effective at suppressing outliers.

Table 5: Statistical analysis of transformed Q projection weights. The neural encoder significantly reduces kurtosis and outliers compared to other transformations.

| Method | Kurtosis | Max Value ($\sigma$) | Outliers ($> 3\sigma$) |
|---|---|---|---|
| Original | 20.48 | 43.57 | 1.94% |
| DCT | 0.46 | 10.54 | 0.50% |
| Random Rotation | 5.42 | 16.77 | 1.86% |
| RHT | 5.44 | 16.66 | 1.86% |
| **Neural Encoder** | **0.00** | **2.70** | **0.00%** |

## E    IMPORTANCE-AWARE WEIGHTING COMPARED TO AWQ AND GPTQ

In this section, we explicitly compare the proposed NWC with existing post-training quantization methods like AWQ and GPTQ. Our method differs in two critical aspects:

- **Discrete importance:** For compact storage, we discretize the scaling factors into a small set of levels (e.g., $K = 4$). These levels can be stored using only trivial overhead (e.g., 2 bits per channel). In contrast, the sensitivity metrics in GPTQ and AWQ are continuous.
- **Importance-augmentation during training:** To address the imbalance in samples across importance levels, we conduct importance-augmented training where each vector is paired with a randomly selected scaling factor.

### E.1    IMPORTANCE-AWARE WEIGHTING IN NWC

Let $\mathbf{X} \in \mathbb{R}^{d \times m}$ denote the layer input corresponding the $m$ calibration data samples. The per-layer Hessian can be obtained by, $\mathbf{H} = \frac{1}{m}\mathbf{X}\mathbf{X}^\top$. We define the importance of the each weight matrix:

$$\mathbf{s} = \frac{\mathbf{\Lambda}}{\frac{1}{d}\sum_{i=1}^{d}\mathbf{\Lambda}_i}, \quad \mathbf{\Lambda} = \mathrm{diag}(\mathbf{H}) \tag{9}$$

The discrete levels $\{\lambda_k\}_{k=1}^{K}$ in Section 4.2 are derived from the empirical entry-wise distribution of $\mathbf{s}$. Specifically, we divide the values of $\mathbf{s}$ into $K$ intervals defined by their quantiles. Letting $B_k$ denote the set of values in the $k$-th interval, we select representative level as $\lambda_k \in B_k$. This approach ensures that the discrete levels effectively capture the density of the importance distribution.

### E.2    ACTIVATION-AWARE SCALING IN AWQ

AWQ's activation-aware scaling is defined as

$$\mathbf{s} = \mathbf{s_X}^{\alpha^*}, \quad \alpha^* = \arg\min_{\alpha \in [0,1]} \mathcal{L}(\mathbf{s_X}^\alpha) \tag{10}$$

where $(\mathbf{s_X})_c = \frac{1}{N}\sum_{i=1}^{N}|\mathbf{X}_{c,i}|$, and $\mathcal{L}(\mathbf{s})$ is quantization objective of Equation (4) in Lin et al. (2024).

This scaling mechanism relies on the empirical assumption that the quantization step size $\Delta = \max(|\mathbf{w}|)$, remains stable even after scaling (i.e., $\Delta' = \max(|\mathbf{s}^\top\mathbf{w}|) \approx \Delta$). As the scaling factor $s$ increases, this assumption breaks down, forcing AWQ to use a heuristic search to find a safe $\alpha$.

As a fully learnable framework, NWC can explicitly incorporate the scale into its loss function, allowing the optimizer to find the most effective compression directly with respect to the importance.

### E.3    SENSITIVITY METRICS IN GPTQ

Drawing from OBQ (Frantar & Alistarh, 2022), GPTQ leverages arbitrary ordering and Cholesky reformulation to define the error sensitivity metrics of each weight column as:

$$\mathbf{s} = \mathrm{diag}(\mathbf{L}^\top)^{-1}, \quad \mathbf{L} = \mathrm{Cholesky}(\mathbf{H}^{-1}). \tag{11}$$

Adopting this as an importance metric, however, is computationally more intensive than simply using the Hessian diagonal $\Lambda$. It requires one additional matrix inversion and one Cholesky decomposition for each Hessian matrix. Empirically, we find that using the diagonal of the Hessian directly is not only sufficient but often yields superior results for training neural compression models.

## F    JUSTIFICATION FOR DIAGONAL HESSIAN APPROXIMATION

In Section 4.2, we employ a diagonal approximation of the Hessian matrix to measure parameter sensitivity. While the full Hessian captures cross-parameter correlations, we justify the adequacy of the diagonal approximation based on the distinct activation characteristics–outlier activations– in LLMs.

LLMs are known to massive outlier activations, where specific feature dimensions possess magnitudes significantly larger than others Sun et al. (2024a); An et al. (2025). Let $\mathbf{x} \in \mathbb{R}^{d_{in}}$ denote the input activation. The Hessian is typically approximated using the expected outer product of the inputs, $H \approx \mathbb{E}[\mathbf{x}\mathbf{x}^\top]$.

Consider a feature dimension $d$ that corresponds to an outlier feature. Mathematically, if the magnitude of this outlier feature is significantly larger than other dimensions (i.e., $|x_d| \gg |x_j|$ for $j \neq d$), the diagonal term $H_{dd}$ dominates the off-diagonal terms:

$$H_{dd} \approx \mathbb{E}[x_d^2] \gg \mathbb{E}[x_d x_j] \approx H_{dj} \tag{12}$$

The Hessian matrix becomes effectively diagonally dominant in the presence of strong outliers. Therefore, the diagonal approximation serves as accurate and computationally efficient proxy for parameter importance.

## G    THE USE OF LARGE LANGUAGE MODELS

We utilized a large language model (LLM) to refine the language and improve clarity in several sections of this paper. However, its use was strictly limited to improving the writing style; the LLM did not contribute to the research ideation or the core scientific content.

