# OpenReview forum: "Neural Weight Compression for Language Models"
_ICLR.cc/2026/Conference — Submitted to ICLR 2026_

### Official Review · Reviewer_CDm5 · 2025-10-22

**Soundness:** 3
**Presentation:** 4
**Contribution:** 2
**Rating:** 4
**Confidence:** 4

**Summary:**

Describes a weight compression approach that uses a trained neural network to assist quantisation. Tensors are split column-wise, and a scale per column is stored to normalise them to unit variance. Each column is also assigned an importance level using a local Hessian of layer output reconstruction error, computed from calibration data. The column is split into blocks of 16 weights, which are transformed using a small _encoder_ residual network, which is conditioned on the importance level, before quantisation and entropy encoding. Decoding uses an entropy decoder, followed by a mirror-image _decoder_ network, also conditioned on importance level, then multiplied by the column scale. This is combined with extant enhancements for incremental quantisation within a layer and across layers.

The paper contributes this specific scheme, the first (that the authors are aware of) to use a single network for compression of all model weights in a large model, and a comparison of its performance against popular data-aware post training quantisation techniques for language and vision transformers, where it performs favourably, especially for high bit-rates $\geq 4$ bits per parameter.

**Strengths:**

The proposed neural weight compression (NWC) scheme is relatively straight-forward and is well-motivated. I find the use of sensitivity as a conditioning signal to the encoder/decoder pair as well as a weighting on the reconstruction error to be an interesting feature. The inclusion of both language and vision transformers in the experimental results is appreciated. Notation is clear throughout, diagrams are very good and plots are clear. The related work section on neural codecs is helpful for positioning this work against previous approaches. A key limitation of hardware efficiency is clearly acknowledged.

**Weaknesses:**

My main concern regarding this work concerns the scientific contribution. Although the method is well-motivated, the paper does not claim direct practical application as decoding speed is reserved for future work. This is reasonable, however I would then expect strong theoretical or empirical results that can guide future work in this space. I do not think the empirical investigation meets this bar, primarily due to a lack of relevant baselines or deeper insight into what is learnt by the trainable network.

Specific concerns:

1. No alternative neural codec baselines are offered. For example, ReALLM (Leconte et al., 2024) is mentioned in related work, and seems applicable, having been tested on LLaMA 2 in the original work.
1. Many design decisions are not ablated. I appreciate the ablations of fitting to actual weight distributions, importance awareness, intra-layer and inter-layer error recovery. However the following also seem relevant:
   - Use of an encoder/decoder network at all. Unless I'm mistaken, the entropy model could be trained using the aux loss, and $g_a$ and $g_s$ could be replaced with the identity function(?) This would test the specific benefit of the learned network.
   - Use of entropy encoding, versus a fixed-length nonlinear quantiser.
   - Channel-wise normalisation, versus tensor-wise.
   - Encoder/Decoder architecture design, depth etc.
1. No comparison against quantisation aware training techniques such as LLM-QAT (Liu et al., 2023), or alternatively any explanation given as to why this comparison is out of scope. A usual reason would be the training time needed to quantise a model, but since this technique involves training the quantisation network and incremental fine-tuning, it is not immediately obvious that this would outperform QAT.
1. No attempt to understand the function that is learnt by the NWC model, beyond the example of Section 3.2 demonstrating the importance of codec fit at high bitrates.
1. Missing details.
   - The entropy model "deep-factorized prior" mentioned in L315 is not sufficiently clear from the reference (Ballé et al., 2016), and similarly for the entropy code itself - is it the arithmetic code from the same paper, Section 6.2?
   - Codec training duration (samples or epochs)? I assume the training dataset size for language models is approximately 8B/16 = 500M examples.
   - Training procedure - presumably using a straight-through estimator, with the noise described in L187? (Perhaps this is just a wording quibble on "typically...").

Minor concerns:

 - I don't believe Equations 2, 3 are a good approximation to the optimal scalar quantiser with a fixed number of codepoints (where the codepoint density should follow the cube root of the pdf) or under entropy coding (where the codepoint density should be uniform).
 - Equation 1 might be clearer with an explicit expectation for $\mathbf{x} \sim$ Dataset.
 - L364 makes the notation slightly unclear. Presumably the combined distortion loss multiplier is $\lambda \cdot \lambda_q$?
 - Table 2 shows an entropy decoding latency of 13.8 ms for a 256x256 tensor, which seems unreasonably bad (I estimate 28 minutes to decode an 8B model). I accept the premise that this work considers hardware acceleration for future work, but using this result to suggest that synthesis is relatively cheap could be misleading.

---

_Leconte, L., Bedin, L., Nguyen, V.M. and Moulines, E., 2024. ReALLM: A general framework for LLM compression and fine-tuning._

_Liu, Z., Oguz, B., Zhao, C., Chang, E., Stock, P., Mehdad, Y., Shi, Y., Krishnamoorthi, R. and Chandra, V., 2023. LLM-QAT: Data-free quantization aware training for large language models._

_Ballé, J., Laparra, V. and Simoncelli, E.P., 2016. End-to-end optimized image compression._

**Questions:**

My questions are included in the weaknesses section.

---

> ### Author Response · Authors · 2025-11-26
>
> Dear Reviewer CDm5,
>
> We greatly value detailed feedback and thoughtful questions, which have helped clarify and enhance the understanding of our work. We provide detailed responses to the questions below.
>
> ----
> ### **W1.** Additional comparison with ReALLM.
> > No alternative neural codec baselines are offered. For example, ReALLM (Leconte et al., 2024) is mentioned in related work, and seems applicable, having been tested on LLaMA 2 in the original work.
>
> We have updated ***Figures 5 and 7*** to include the results of ReALLM, along with the corresponding discussions in the revised manuscript.
>
> We observe that NWC achieves comparable or better performance to ReALLM, even without computationally intensive finetuning—ReALLM requires approximately $40\times$ more time to update low-rank components via end-to-end fine-tuning. Crucially, in data-free settings, NWC is shown to exhibit significantly superior performance.
>
> ----
> ### **W2.** Additional ablation studies.
> > Many design decisions are not ablated. I appreciate the ablations of fitting to actual weight distributions, importance awareness, intra-layer and inter-layer error recovery. However the following also seem relevant:
>
> We thank the reviewer for suggesting the relevant ablation studies. We have added more ablation studies and in ***Figure 9, 13, and Table 4*** of the revised version, in addition to the ablations we already provide in Section 5.5.
>
>
> > 1. Use of an encoder/decoder network at all. Unless I'm mistaken, the entropy model could be trained using the aux loss, and and could be replaced with the identity function(?) This would test the specific benefit of the learned network.
>
>
> We investigated the benefit of the learnable transformation networks ($g_a$/$g_s$) by comparing them against identity mappings (trained with only auxiliary and rate loss).
>
> The results suggest that using the learnable networks ($g_a$/$g_s$) leads to a substantial reduction in perplexity (PPL).
>
> |En/Decoder | Bits | Wiki2 | C4 |
> |- |-|-|-|
> | Base model| 16 | 6.1 | 8.9 |
> |Identity|2.12 | 27.8 | 33.2 |
> |$g_a$/$g_s$ | 2.13 | 14.5 | 20.7 |
>
>
> > 2. Use of entropy encoding, versus a fixed-length nonlinear quantiser.
>
> We compared the performance of our codec's entropy model against a fixed-length non-linear scalar quantizer (SQ), and Vector Quantizer (VQ) module to assess the necessity of entropy coding in *Figure 13* in the revised version.
>
> The results clearly demonstrate the critical role of the entropy model.
>
> | Quantizer | Bits | Wiki2 | C4 |
> |- |-|-|-|
> | Base model| 16 | 6.14 | 8.88 |
> |VQ| 4.50 | 6.89 | 9.86 |
> |Entropy coder| 4.29 | **6.40** |**9.19** |
> |SQ| 3.00 | 36.45 | 50.70|
> |VQ| 3.00 | 12.09 | 16.50 |
> |Entropy coder| 3.01 | **7.37** | **10.54** |
> |SQ| 2 | Inf | Inf |
> |VQ| 2.01 | $9.85 \times 10^4$ | $6.13 \times 10^4$ |
> |Entropy coder| 2.16 | **14.83** | **19.53**|
>
>
> > 3. Channel-wise normalization, versus tensor-wise.
>
> We compare the performance when normalization is applied at different levels: globally (across the entire model weights), tensor-wise (per-layer), and channel-wise. The results consistently show that a finer granularity (i.e., channel-wise) leads to lower perplexity, requiring only a small overhead to store the necessary normalization factors.
>
> | Quantiser | Bits | Wiki2 | C4 |
> |-|-|-|-|
> | Base model| 16 | 6.14 | 8.88 |
> |Global|2.24|13.62|18.18|
> |Tensor-wise|2.28|10.69|14.88|
> |Channel-wise|2.29|10.26|14.08|
>
> > 4. Encoder/Decoder architecture design, depth etc.
>
> These experiments are still underway due to a large search space. We will add these results as soon as they become available.

---

> ### Author Response · Authors · 2025-11-26
>
> ### **W3.** Additional comparison with QAT
> > No comparison against quantisation aware training techniques such as LLM-QAT (Liu et al., 2023), or alternatively any explanation given as to why this comparison is out of scope. A usual reason would be the training time needed to quantise a model, but since this technique involves training the quantisation network and incremental fine-tuning, it is not immediately obvious that this would outperform QAT.
>
> Thank you for pointing this out. In the revised manuscript, we have added clarifications on why we focused on comparisons with PTQ (see Appendix B.3). The main reason is that QAT methods, e.g., LLM-QAT, ***require significantly more computation*** than ours. Specifically, training NWC including finetuning requires only 12 hours on an NVIDIA RTX 6000 Ada, whereas LLM-QAT demands approximately 280 hours on an A100 80G.  Furthermore, QAT relies on the full ***pre-training corpus or extensive synthetic data***, whereas NWC requires only the small calibration set.
>
> Nevertheless, we provide a direct comparison with LLM-QAT and BitDistiller [1] in **Figure 12** in the received version. *NWC outperforms QAT* methods in both perplexity and zero-shot accuracy (ARC-c, PIQA, HellaSwag, WinoGrande)
>
> | Quantizer | Bits | WikiText-2 | Zero-shot Acc(%)|
> |-|-|-|-|
> |Base model| 16 | 5.47 | 61.9|
> |LLM-QAT | 3.25 | 6.02 | 56.46 |
> |BitDistiller | 3.25 |  5.97 | 57.12 |
> |NWC (Ours) | 3.17 | **5.68** | **60.6** |
> |LLM-QAT | 2.25 | 9.30 | 45.64 |
> |BitDistiller | 2.25 |  8.08 | 49.18 |
> |NWC (Ours) | 2.31 | **6.42** | **57.9** |
>
>
> ----
> ### **W4.**  Further investigation into the learned function
>
> > No attempt to understand the function that is learnt by the NWC model, beyond the example of Section 3.2 demonstrating the importance of codec fit at high bitrates.
>
> We deeply appreciate this suggestion. To address this, we investigated the distributional properties of the latent representations produced by the NWC encoder. We found that the NWC encoder functions as a highly effective ***outlier suppressor***, significantly outperforming traditional linear transformations used in quantization literature (e.g., Hadamard or FFT).
>
> As detailed in the newly added **Appendix D**, we compared the kurtosis and outlier statistics ($>3\sigma$) of the original weights ($W$) against those transformed by FFT ($W_{\text{fft}}$), Hadamard ($W_{\text{had}}$), and our NWC encoder ($W_{\text{encoded}}$).
>
> | Tensor | Kurtosis | Max Sigma | Outliers (>3σ) |
>  | :--- | :--- | :--- | :--- |
>  | $W$ (Original) | 20.48 | 43.57 | 1.94% |
>  | $W_{\text{fft}}$ | 1.10 | 11.63 | 0.82% |
>  | $W_{\text{had}}$ | 8.08 | 24.92 | 1.77% |
> | **$W_{\text{encoded}}$** | **0.00** | **2.69** | **0.00%** |

---

> ### Author Response · Authors · 2025-11-26
>
> ### **W5.** Other details.
>
> We sincerely thank the reviewer for the detailed feedback regarding the implementation specifics. We have incorporated these missing details into the revised manuscript to ensure clarity and reproducibility.
>
> > The entropy model “deep-factorized prior" mentioned in L315 is not sufficiently clear from the reference (Ballé et al., 2016), and similarly for the entropy code itself - is it the arithmetic code from the same paper, Section 6.2?
>
> **Entropy model and coding.**
>
> We adopt the *non-parametric, fully factorized density model* introduced by [2]. This model assumes statistical independence among latent elements and approximates their marginal distributions using piecewise linear functions. Our implementation follows this mechanism, learning the cumulative distribution functions (CDFs) for these factorized marginals.
>
> For entropy coding, we utilize *arithmetic coding* based on these learned probability estimates, which is consistent with the framework described in Section 6.2 of the reference.
>
> > Codec training duration (samples or epochs)? I assume the training dataset size for language models is approximately 8B/16 = 500M examples.
>
> **Codec training duration and dataset size.**
>
> The codec was trained for 60 epochs. The training dataset consists of approximately 436 million examples, with a validation set of 64,000 samples. For computational efficiency during random sampling, we grouped samples into batches of 128.
>
> > Training procedure - presumably using a straight-through estimator, with the noise described in L187? (Perhaps this is just a wording quibble on "typically...").
>
> **Gradient estimation.**
>
> We do not use the straight-through estimator during training. Instead, we simulate quantization by applying additive uniform noise, consistent with the method described in L187 and [2].
>
> ---
> ### **Minor concerns**
> > I don't believe Equations 2, 3 are a good approximation to the optimal scalar quantiser with a fixed number of codepoints (where the codepoint density should follow the cube root of the pdf) or under entropy coding (where the codepoint density should be uniform).
>
> **About Equations 2, 3**
>
> Thank you for pointing this out. Indeed, this is our mistake—in the revised version, we have corrected the equations and corresponding figures, using the Panter-Dite companders (with cube root PDFs).
>
> We note that, however, ***our main observation*** remains unchanged: the Gaussian-optimized quantizer still suffers from performance degradation on heavy-tailed weights. We update the section 3.2 to reflect the Panter-Dite condition, ensuring theoretical precision while maintaining our original conclusion.
>
> > Equation 1 might be clearer with an explicit expectation for Dataset.
>
> **About Equations 1.**
>
> We have revised Equation 1 in the updated manuscript to explicitly include the expectation over the dataset, $\mathbb{E}_{x \sim \mathcal{D}}$, ensuring mathematical precision.
>
> > L364 makes the notation slightly unclear. Presumably the combined distortion loss multiplier is ?
>
> **Notation at L364.**
>
> We initially simplified the notation for brevity, but to ensure explicitness, we have revised  L364 to clearly define the combined distortion loss multiplier.
>
> > Table 2 shows an entropy decoding latency of 13.8 ms for a 256x256 tensor, which seems unreasonably bad (I estimate 28 minutes to decode an 8B model). I accept the premise that this work considers hardware acceleration for future work, but using this result to suggest that synthesis is relatively cheap could be misleading.
>
> **Decoding Latency in Table 2.**
>
> In the original manuscript, the value reported in Table 2 was measured using a CPU-based implementation of arithmetic coding, which was heavily bottlenecked by the data communication overhead between the CPU and GPU.
>
> We have re-evaluated the decoding latency assuming a GPU-based parallel implementation of arithmetic coding, achieving a reduction of over ***95%***. Consequently, decoding the full LLaMA-3-8B model takes only tens of seconds. We have updated **Table 3** in the revised manuscript with these results.
>
> ----
> [1] Du et al., BitDistiller: Unleashing the Potential of Sub-4-Bit LLMs via Self-Distillation, 2024
>
> [2] Ballé et al., End-to-End Optimized Image Compression. ICLR 2017.
>
> [3] Panter and Dite, Quantization distortion in pulse-count modulation with nonuniform spacing of levels, 1951.

---

### Official Review · Reviewer_XFXJ · 2025-10-24

**Soundness:** 4
**Presentation:** 4
**Contribution:** 2
**Rating:** 4
**Confidence:** 4

**Summary:**

The paper proposed to compress neural network weights using another neural network.
Achieves competitive performance in 4-6 bits per weight range.

But it is not practical for deployment.

**Strengths:**

Presentation of the paper is quite clear.
Also optimization procedure is good and well written.
Results are strong for both language and vision models in 4-6 bit range.

**Weaknesses:**

The main problem is a lack of proper inference-time support and the fact that the method is not practical for deployment.
If we say that compression methods without practical deployment are interesting, then it opens doors for a plethora of methods, such as compressing quantized weights via 7zip (this works very well, but nobody does that for obvious reasons).

At least the implications of decompression during inference should be properly discussed in the paper. I can imagine doing heavy decompression in a big batch scenario, but I cannot imagine doing it in a local deployment with batch 1 LLM decoding.

The secondary problem is that when the network is compressed, it cannot be easily fine-tuned (I do not mean LoRA, but full model fine-tuning).

Also, section 4.2 talks about the Hessian and then just drops back to simple Wanda input activation scaling. I think authors should be honest and not even mention Hessian here.

**Questions:**

- What is inference time now for vision models? (I understand that the proposed solution is impractical for LLM decoding, but if each layer needs to be loaded once, as in vision models, we might get some meaningful tradeoffs)
- Figure 3 and the details of the compression network are unclear. What are the quantizer, AE, and AD in Figure 3?

---

> ### Author Response · Authors · 2025-11-26
>
> Dear Reviewer XFXJ,
>
> We thank you for your constructive suggestions, as well as the effort you put into reviewing our work. Below, we provide responses to the weaknesses and questions you mentioned.
>
> ----
> ### **Remark.** Updates on the decoding latency
> Before addressing individual concerns, we note that we have updated the wall-clock latency table (Table 3). The main changes are twofold:
>
> - GPU-based entropy en-/decoding: We have adopted GPU-based entropy decoding (instead of CPU-based, in the initial version), which saves ***more than 95%*** of the decoding time.
>
> - Baselines: We have added the wall clock latencies of several baselines, for an easier comparison.
> We sincerely believe that this update will help better resolve some of your concerns.
>
> ----
> ### **W1.** About inference-time deployment
> > The main problem is a lack of proper inference-time support and the fact that the method is not practical for deployment. If we say that compression methods without practical deployment are interesting, then it opens doors for a plethora of methods, such as compressing quantized weights via 7zip (this works very well, but nobody does that for obvious reasons).
>
> We respectfully disagree to this point. We strongly believe that our work offers significant value beyond the constraints of immediate deployment, for the following reasons.
>
> The benefits of the weight compression framework are ***not limited to in-GPU acceleration***. Rather, there are critical applications where minimizing bit-rate is more important than pure decoding latency—such as bandwidth reduction in inter-chip communication, distributed training relying on periodic weight exchange, and storage optimization for massive personalized weights.
>
> NWC critically differs from the *Quantize-then-7zip* paradigm by employing *learnable nonlinear transform coding.* As demonstrated in **Appendix D.3**, this nonlinear transformation effectively handles heavy-tailed distributions, leading to substantial performance gains as shown in **Table 5**.
>
>
> > At least the implications of decompression during inference should be properly discussed in the paper. I can imagine doing heavy decompression in a big batch scenario, but I cannot imagine doing it in a local deployment with batch 1 LLM decoding.
>
> Thank you for this suggestion. Following this comment, we have added relevant discussions in **Section 6.** In particular, if we consider in-GPU communication, our method indeed goes better with large-scale serving scenarios with bigger batch sizes, where the decompression overhead is effectively amortized across concurrent requests, rather than small-batch cases.
>
> ----
> ### **W2.** About fine-tuning compressed weight
> > The secondary problem is that when the network is compressed, it cannot be easily fine-tuned (I do not mean LoRA, but full model fine-tuning).
>
> Applying full model fine-tuning directly to a compressed representation is indeed a non-trivial challenge, a limitation shared with most SOTA Post-Training Quantization (PTQ) methods. While Quantization-Aware Training (QAT) attempts to address this, it often incurs prohibitive computational costs, and as demonstrated in our comparison with QAT in **Figure 12**, does not necessarily yield superior performance.
>
> However, we would like to highlight that recent research in neural compression is actively addressing this through ***latent optimization*** techniques [1, 2, 3, 4]. This capability suggests that the neural compression framework holds great potential for effective adaptation as a future direction.
>
> Finally, we note that our method offers a practical advantage due to its fast encoding (**Table 3**) and  generalizability. One can simply decode the weights, perform standard fine-tuning, and then re-encode the updated weights using the same pre-trained codec. This approach does not require any further training of the encoder-decoder pairs, making this iterative process efficient.

---

> ### Author Response · Authors · 2025-11-26
>
> ### **W3.** Suggestions on Hessian-based derivation of the method
> > Also, section 4.2 talks about the Hessian and then just drops back to simple Wanda input activation scaling. I think authors should be honest and not even mention Hessian here.
>
> We respectfully disagree with the reviewer’s claim that making connections to Hessian should be viewed as ***“dishonest.”***
>
> As a matter of fact, the diagonal approximation has already appeared in OBD and SparseGPT, with SparseGPT already noting that the *Layerwise Hessian* can be approximated with input activation scales. In fact, even the original Wanda paper explicitly notes the connection to these works and the Hessian (see “remarks” of section 3). Thus, we view Hessian-centric description as a way to derive these scores from the first principles, making the manuscript more self-contained.
>
> That said, we have revised the manuscript to make it clearer that (1) this approximation has appeared in Wanda, and (2) we are not claiming this work as the first to do such derivations.
>
> ----
> ### **Q1.** Inference time for vision models
> > What is inference time now for vision models? (I understand that the proposed solution is impractical for LLM decoding, but if each layer needs to be loaded once, as in vision models, we might get some meaningful tradeoffs)
>
> Thank you for this insightful comment. For CLIP-Base model, the total decoding latency is 34.9 ms, which is slightly but not critically slower than baselines, such as QuIP# (which requires 26.5 ms). Thus, as the reviewer suggested, we believe that the tradeoff on vision models may lead to a meaningful consequence, under in-GPU communication scenarios.
>
> ----
> ### **Q2.** Details on notations
> > Figure 3 and the details of the compression network are unclear. What are the quantizer, AE, and AD in Figure 3?
>
> We have followed the convention in the literature (e.g., [5, 6, 7]) where quantizer denotes the rounding operator of latent representation, AE denotes Arithmetic encoding ,and AD denotes arithmetic decoding. We have added these descriptions to the Figure 3 captions.
>
> ----
>
> [1] Campos et al., Content Adaptive Optimization for Neural Image Compression, 2019.
>
> [2] Yang et al., Improving Inference for Neural Image Compression, NeurIPS 2020.
>
> [3] Gao et al., Flexible neural image compression via code editing, NeurIPS 2022.
>
> [4] Perugachi-Diaz et al., Robustly overfitting latents for flexible neural image compression, NeurIPS 2024.
>
> [5] Ballé et al., End-to-End Optimized Image Compression. ICLR 2017.
>
> [6] Richard Schwarz et al., Modality-Agnostic Variational Compression of Implicit Neural Representations, ICLM 2023.
>
> [7] Liu et al., Rate-Distortion-Cognition Controllable Versatile Neural Image Compression, ECCV 2024.

---

> > ### Comment · Reviewer_XFXJ · 2025-11-27
> >
> > Thank you for the response, but I am still confused about decoding latency/implementation.
> >
> > In appendix, you say: "The decoding latency of NWC is estimated by assuming a GPU-based parallel implementation of arithmetic coding"
> > Does this mean that this implementation does not exist and you just estimated the speed using some guess? Or do you have an actual fast decoding implementation?
> >
> > "For CLIP-Base model, the total decoding latency is 34.9 ms, which is slightly but not critically slower than baselines, such as QuIP# (which requires 26.5 ms). Thus, as the reviewer suggested, we believe that the tradeoff on vision models may lead to a meaningful consequence, under in-GPU communication scenarios."
> >
> > I need more information here. What is the batch/image size? What is the speed for the non-compressed model?

---

> > > ### Author Response · Authors · 2025-11-28
> > >
> > > Thank you very much for the prompt reply!
> > > We appreciate your continuing efforts to provide us feedback.
> > >
> > > > In Appendix, you say: “The decoding latency of NWC is estimated by assuming a GPU-based parallel implementation of arithmetic coding” Does this mean that this implementation does not exist and you just estimated the speed using some guess? Or do you have an actual fast decoding implementation?
> > >
> > > Sorry for the confusion. The results are based on actual measurements, not guesses. Specifically, we measured the synthesis step directly on the GPU, and for entropy decoding, we benchmarked the latency using NVIDIA’s nvCOMP library (ANS implementation) on the exact compressed data size.
> > > We used the word 'estimated' because the two steps were benchmarked separately rather than running as a single fused kernel. We have revised Appendix B.4 to clarify this point.
> > >
> > > > I need more information here. What is the batch/image size? What is the speed for the non-compressed model?
> > >
> > > The reported decoding latency denotes the time required to decompress the model weights to the original form---i.e., it excludes the time required to transmit the weight from the memory to the computing units. Thus, the speed for the non-compressed model is 0.
> > >
> > > We guess that the reviewer is interested in a more comprehensive analysis, including the time to transport the weight and the throughput of the decoded models. To answer this question, we have added more information to the table 3. In short:
> > >
> > > - The throughput of the decoded models is identical among all models (as they are all in FP32 whenever decoded). The total latency for a 128-sample batch (assuming 224x224 resolution), is 317.84 ms.
> > >
> > > - The transport latency of the uncompressed CLIP-Base model depends on the data transmission medium used; assuming a PCIe 4.0 x16 interface, the latency is 108.0 ms, while that of the (2.3bit- )compressed model is 43.6 ms. The combined transport and decoding latency is 43.6 + 34.9 = 78.5 ms, which is lower than the uncompressed transport latency.

---

### Official Review · Reviewer_H8K4 · 2025-11-01

**Soundness:** 2
**Presentation:** 3
**Contribution:** 2
**Rating:** 2
**Confidence:** 3

**Summary:**

This paper introduces Neural Weight Compression (NWC), an autoencoder-based neural codec that learns compact, compressible representations of pretrained LLM weights. Unlike conventional quantization approaches that rely on handcrafted transformations such as Hadamard transforms or per-channel scaling, NWC directly learns nonlinear mappings from the weight data itself. To address the challenges of heterogeneous tensor structures and varying parameter importance, NWC employs column-wise tensor chunking and normalization, coupled with an importance-aware loss function that assigns adaptive quality levels to different weight chunks based on estimated Hessian diagonals. During inference, an error compensation mechanism further enhances reconstruction quality by integrating intra-layer LDL-based feedback with inter-layer recovery fine-tuning. Extensive experiments on several LLaMA models and vision encoders have been conducted.

**Strengths:**

1. The paper is clearly structured and technically detailed, with informative figures.

2. This paper addresses an important and increasingly relevant problem: learned compression of LLMs for on-device or bandwidth-limited deployment.

3. This paper includes ablations and qualitative visualizations that clarify the empirical behavior of the codec.

**Weaknesses:**

1. The paper exhibits limited novelty compared to recent neural codec frameworks. Similar learned compression paradigms involving codebook-based quantization and additive composition have already been introduced in AQLM [1], LCQ [2], and QuIP# [3].

2. The work also lacks theoretical grounding for its importance-aware loss. While the use of Hessian-weighted objectives follows the sensitivity-based quantization formulation seen in GPTQ [4], the paper does not justify the adequacy of diagonal approximations in highly non-convex LLM parameter spaces, nor provide convergence or variance analysis to support its objective function.

3. The experimental rigor is weak, with marginal improvements typically under 0.3 perplexity or 1% accuracy. These results are not accompanied by confidence intervals or variance statistics. Furthermore, the evaluation focuses primarily on small LLaMA-2 and LLaMA-3 models, limiting generalization to larger architectures such as Qwen or Mixtral.

4. The paper fails to address deployment overhead. The runtime cost of the decoder is unreported, and comparisons with QA-LoRA [7] and QTIP [8] omit critical latency and throughput metrics. Given that reconstruction is required at inference time, the actual efficiency gains remain questionable.

5. Another key shortcoming is the absence of component-wise ablation. The individual contributions of chunking, importance weighting, and error compensation are not disentangled, leaving the source of improvement unclear.

6. The paper omits several strong baselines, including BitDistiller [10], SVD-LLM [5], LoSparse [6], and UniCodec [9], all of which represent recent advances in model compression or quantization.

References

[1] Egiazarian et al., Extreme Compression of Large Language Models via Additive Quantization (AQLM), 2024.

[2] Cai et al., LCQ: Low-Rank Codebook Based Quantization for Large Language Models, 2024.

[3] Tseng et al., QuIP#: Even Better LLM Quantization with Hadamard Incoherence and Lattice Codebooks, 2024.

[4] Frantar et al., GPTQ: Accurate Post-Training Quantization for Generative Pre-Trained Transformers, 2023.

[5] Wang et al., SVD-LLM: Truncation-Aware Singular Value Decomposition for LLM Compression, 2024.

[6] Li et al., LoSparse: Structured Compression of Large Language Models Based on Low-Rank and Sparse Approximation, 2023.

[7] Xu et al., QA-LoRA: Quantization-Aware Low-Rank Adaptation of Large Language Models, 2024.

[8] Tseng et al., QTIP: Quantization with Task-Informed Priors, 2024.

[9] Jiang et al., UniCodec: Unified Audio Codec with Single Domain-Adaptive Codebook, 2025.

[10] Du et al., BitDistiller: Unleashing the Potential of Sub-4-Bit LLMs via Self-Distillation, 2024.

**Questions:**

1. How does the proposed framework fundamentally differ from recent neural codec approaches such as AQLM [1], LCQ [2], and QuIP# [3], beyond minor variations in tensor organization or weighting strategy?

2. Can the authors justify why a diagonal Hessian approximation is theoretically adequate for measuring parameter importance in highly non-convex LLM parameter spaces, as opposed to more expressive curvature models?

3. Are the reported sub-1 % accuracy and < 0.3 perplexity gains statistically significant, and can the authors provide confidence intervals or multi-seed variance analyses to substantiate these results?

4. What is the measured runtime and latency overhead introduced by the decoder, and do the claimed memory savings still hold once inference-time reconstruction costs are included relative to AWQ [7] and QTIP [8]?

5. Could the authors present an ablation isolating the effects of chunking, importance weighting, and error compensation to clarify which components contribute most to performance?

6. How does the proposed method compare with BitDistiller [10], SVD-LLM [5], LoSparse [6], and UniCodec [9]?

---

> ### Author Response · Authors · 2025-11-26
>
> Dear Reviewer H8K4,
>
> We sincerely appreciate your valuable feedback, which has enriched our experiments and baselines. Below, we address your questions in detail.
>
> ----
> ### **W1.** Clarifications on the difference of the proposed framework and AQLM/LCQ/QuIP#
> > The paper exhibits limited novelty compared to recent neural codec frameworks. Similar learned compression paradigms involving codebook-based quantization and additive composition have already been introduced in AQLM [1], LCQ [2], and QuIP# [3].
>
> > How does the proposed framework fundamentally differ from recent neural codec approaches such as AQLM [1], LCQ [2], and QuIP# [3], beyond minor variations in tensor organization or weighting strategy?
>
> Our framework fundamentally differs from these approaches in its structural paradigm: (1) Nonlinear Transformation, and (2) Density estimation-based entropy coding. We have clarified this in **Section 2** in our revised version.
>
> **Density estimation-based entropy coding**: Unlike codebook-based methods, our compression is performed via entropy coding. Instead of searching for fixed discrete indices, our method learns the underlying probability distribution of the latent representation, enabling precise variable-rate compression via arithmetic coding. The effectiveness of this entropy coding strategy is demonstrated in the **table 4** of the revised manuscript.
>
> **Nonlinear Transform**: Our approach employs a non-linear neural autoencoder (consisting of a learnable encoder $g_a$ and decoder $g_s$), in contrast to the linear scaling used in AQLM and LCQ, or the Hadamard-based randomized linear operations in QuIP#. This non-linearity allows the codec to effectively capture the heavy-tailed nature of the weight distribution, as discussed in **Section D.2** of the revised manuscript.
>
> ----
> ### **W2.** Additional justification for Hessian diagonal approximations
> > The work also lacks theoretical grounding for its importance-aware loss. While the use of Hessian-weighted objectives follows the sensitivity-based quantization formulation seen in GPTQ [4], the paper does not justify the adequacy of diagonal approximations in highly non-convex LLM parameter spaces, nor provide convergence or variance analysis to support its objective function.
>
> > Can the authors justify why a diagonal Hessian approximation is theoretically adequate for measuring parameter importance in highly non-convex LLM parameter spaces, as opposed to more expressive curvature models?
>
> The diagonal approximation of the Hessian is accurate for LMs, essentially due to the **outlier activations** that empirically emerge in the models. In a nutshell,  we have $H \approx \mathbb{E}[\mathbf{x}\mathbf{x}^\top]$. If a specific feature dimension $d$ corresponds to an outlier feature, i.e., $|x_d| \gg |x_j|$ for $j \neq d$, the diagonal term $H_{dd} \approx \mathbb{E}[x_d^2]$ becomes the dominant term, significantly outweighing the off-diagonal cross-terms $H_{dj} \approx \mathbb{E}[x_d x_j]$. Therefore, in the presence of such strong outliers characteristic, diagonal approximation becomes increasingly accurate
>
> We have added these discussions in **Appendix F.**
>
> ----
> ### **W3.** Additional experiment result (Qwen/Mixtral, Multi-seed variance)
> > The experimental rigor is weak, with marginal improvements typically under 0.3 perplexity or 1% accuracy. These results are not accompanied by confidence intervals or variance statistics. Furthermore, the evaluation focuses primarily on small LLaMA-2 and LLaMA-3 models, limiting generalization to larger architectures such as Qwen or Mixtral.
>
> > Are the reported sub-1 % accuracy and < 0.3 perplexity gains statistically significant, and can the authors provide confidence intervals or multi-seed variance analyses to substantiate these results?
>
> As suggested, we have trained and evaluated our codec using three different random seeds. The observed standard deviation across these runs was minimal (range < 0.004 in C4 perplexity and < 0.3% in MMLU accuracy), confirming that our reported performance gains are statistically reliable.
>
> Furthermore, we have demonstrated generalization by extending our experiments to larger architectures, specifically Mixtral with 56B parameters (8x7B-v0.1) and Qwen3-30B-A3B. These new results can be found in **Figure 6** of the revised manuscript.

---

> ### Author Response · Authors · 2025-11-26
>
> ### **W4Q4.** Explicit decoding overhead comparison against baselines
> > The paper fails to address deployment overhead. The runtime cost of the decoder is unreported, and comparisons with QA-LoRA [7] and QTIP [8] omit critical latency and throughput metrics. Given that reconstruction is required at inference time, the actual efficiency gains remain questionable.
>
> > What is the measured runtime and latency overhead introduced by the decoder, and do the claimed memory savings still hold once inference-time reconstruction costs are included relative to AWQ [7] and QTIP [8]?
>
> We have addressed your concern by providing a comparison of decoding time against baselines (GPTQ, QuIP#, and QTIP) in **Table 3** of the revised manuscript. Additionally, we update the value of NWC latency to reflect implementation utilizing GPU-based entropy encoding/decoding, which has reduced the latency to approximately 1/20 of the previous CPU-based version. As shown in the table, although NWC incurs overhead compared to the baselines ($\sim$30% increase relative to QuIP#), the gap is not excessively large.
>
> Furthermore, we would like to emphasize that low latency decoding may not be a critical constraint for the potential applications of NWC, such as bandwidth reduction in inter-chip communication, periodic weight exchange in distributed training, and storage optimization for massive personalized weights.
>
> ----
> ### **W5.** Clarifications on component-wise ablations
> > Another key shortcoming is the absence of component-wise ablation. The individual contributions of chunking, importance weighting, and error compensation are not disentangled, leaving the source of improvement unclear.
>
> > Could the authors present an ablation isolating the effects of chunking, importance weighting, and error compensation to clarify which components contribute most to performance?
>
> We clarify that we already provide component-wise ablations in Figures 11, 12, and 13 of the manuscript. The results demonstrate that each component consistently contributes to performance improvement. Specifically, importance weighting plays the most critical role in the low-bit regime, while error compensation yields consistent performance gains across all bitrates.
>
> Furthermore, we added additional ablation about encoder/decoder network and entropy model in **Figure 13 and Table 4.**
>
> ----
> ### **W6.** Additional comparison (QAT, SVD-based)
> > The paper omits several strong baselines, including BitDistiller [10], SVD-LLM [5], LoSparse [6], and UniCodec [9], all of which represent recent advances in model compression or quantization.
>
> > How does the proposed method compare with BitDistiller [10], SVD-LLM [5], LoSparse [6], and UniCodec [9]?
>
> We have incorporated comparative results against BitDistiller, LLM-QAT, and SVD-LLM into **Figure 10** of the revised manuscript.
>
> The results clearly demonstrate that our method exhibits superior rate-distortion trade-off compared to these baselines.
>
> ----
>
> [1] Sagun et al., Empirical Analysis of the Hessian of Over-Parametrized Neural Networks, 2017.
>
> [2] Yao et al., AdaHessian: An Adaptive Second Order Optimizer for Machine Learning, AAAI 2021.
>
> [3] An et al., Systematic Outliers in Large Language Models, ICLR 2025.
>
> [4] Sun et al., Massive Activations in Large Language Models, COLM 2024.

---

### Official Review · Reviewer_L5q5 · 2025-11-01

**Soundness:** 2
**Presentation:** 2
**Contribution:** 2
**Rating:** 2
**Confidence:** 4

**Summary:**

Neural Weight Compression (NWC) is a learning-based framework for compactly representing pretrained model weights through neural encoders and decoders. Rather than depending on fixed quantization heuristics, NWC learns how to compress weight tensors directly, capturing nonlinear relationships across parameters. To manage the diversity of tensor shapes and parameter sensitivities, the method organizes weights into normalized column chunks and optimizes them using an importance-weighted objective that emphasizes more influential components. At inference time, NWC refines the reconstructed weights with a lightweight correction step that leverages Hessian-guided updates to reduce residual error. Evaluations on both language and vision architectures, including LLaMA models, show that NWC delivers strong compression efficiency and accuracy retention, particularly in the 4–6 bit range, across calibration-free and data-driven settings.

**Strengths:**

The idea of learning nonlinear transforms directly from weight tensors is timely and clearly explained.
The training and inference pipeline (Fig. 3) is straightforward to follow, and implementation details are thorough.

The authors provide ablations on the key design choices. The inclusion of both calibration-based and data-free settings adds credibility.

Empirical results show consistent improvements at mid-range bitrates (4–6 bits), particularly on LLaMA-3-8B and the CLIP/SigLIP transfers.
These results suggest the codec captures some general statistical structure of model weights rather than overfitting to one backbone.

**Weaknesses:**

1. The idea of a neural autoencoder trained on weight tensors extends existing work on learned compression [5] and weight quantization [1, 2].
The “importance-aware” weighting resembles the activation-aware scaling in AWQ [2] and the sensitivity metrics in GPTQ [1].
A more precise comparison of these formulations would help clarify the novelty of this paper.

2. Section 3.2 argues that learned codecs outperform handcrafted ones on heavy-tailed distributions, but this remains a qualitative observation.
Without quantitative analysis, such as a per-layer error statistics or rate–distortion modeling, the reader is left unsure why NWC improves mainly in the 4–6 bit regime.

3. Table 2 shows that decoding is dominated by entropy-decoding cost (~14 ms per 256×256 tensor on RTX 6000).
Since the method requires an additional decoding network, the real deployment gains relative to lightweight quantizers such as GPTQ [1] or QuIP [3] are unclear. A direct wall-clock comparison or FLOP breakdown would be valuable.

4. The codec must be fine-tuned for each architecture.
It is unclear how much data and computing are needed for these adaptations.
Reporting training cost or sample requirements would help assess scalability.

5. The main experiments use LLaMA-2 and LLaMA-3, but newer models such as LLaMA-4 and Qwen-3 are already available and would provide a stronger test of generality.

References

[1] E. Frantar et al. GPTQ: Accurate Post-Training Quantization for Generative Pre-Trained Transformers. ICLR 2023.

[2] C.-H. Lin et al. AWQ: Activation-Aware Weight Quantization for On-Device LLM Compression. MLSys 2024.

[3] J. Chee et al. QuIP: 2-Bit Quantization of Large Language Models with Guarantees. NeurIPS 2023.

[4] V. Egiazarian et al. Extreme Compression of Large Language Models via Additive Quantization (AQLM). ICML 2024.

[5] J. Ballé, V. Laparra & E. P. Simoncelli. End-to-End Optimized Image Compression. ICLR 2017.

**Questions:**

1. How does the importance-aware loss differ in practice from activation-aware scaling [2] or Hessian-weighted objectives [1]?

2. What is the total encoding/decoding time compared to GPTQ [1] or QuIP [3] on the same GPU?

3. Does the codec trained on one model (e.g., LLaMA-3-8B) generalize to another without retraining?

4. Could the authors provide per-layer error curves or singular-value spectra to substantiate the claim that heavy-tailed distributions benefit most?

5. Have you considered hybrid schemes combining NWC at mid-bitrates with analytical quantizers at extreme compression levels?

---

> ### Author Response · Authors · 2025-11-26
>
> Dear Reviewer L5q5,
>
> Thank you for your constructive suggestions. Below, we provide responses to the weaknesses and questions you mentioned.
>
> ----
>
> ### **W1Q1.** Clarifications on the difference of the proposed importance-aware weighting and AWQ/GPTQ.
> > The idea of a neural autoencoder trained on weight tensors extends existing work on learned compression [5] and weight quantization [1, 2]. The “importance-aware” weighting resembles the activation-aware scaling in AWQ [2] and the sensitivity metrics in GPTQ [1]. A more precise comparison of these formulations would help clarify the novelty of this paper.
>
> > How does the importance-aware loss differ in practice from activation-aware scaling [2] or Hessian-weighted objectives [1]?
>
> Thank you for the suggestion. To clarify the novelty of our approach, we have explicitly clarified the comparison between our method, AWQ, and GPTQ in the **Appendix E** of our revised manuscript.
> In a nutshell, the proposed importance-aware methods differ from AWQ and GPTQ in two critical aspects:
> - **Discrete importance.** For the sake of compact storage of importance levels, we discretize the scaling factors in 2^2 = 4 levels. The levels can be stored using only two bits per channel, introducing very little overhead. In contrast, sensitivity metrics of GPTQ and AWQ are continuous, and induce more bit overheads.
> - **Importance-augmentation during training.** To address the imbalance in the number of samples with each importance level, we conduct an importance-augmented training where we pair each vector with a randomly selected scaling factor. As demonstrated in Figure 12, this helps boost the performance of the trained codec. In contrast, AWQ and GPTQ rely on static sensitivity metrics for the given calibration set, at inference. They utilize importance solely as a fixed constraint to adjust quantization grids or weight updates.
>
> ----
>
> ### **W2Q4.** Additional Analyses (Error statistics, Spectral Analyses)
> > Section 3.2 argues that learned codecs outperform handcrafted ones on heavy-tailed distributions, but this remains a qualitative observation. Without quantitative analysis, such as a per-layer error statistics or rate–distortion modeling, the reader is left unsure why NWC improves mainly in the 4–6 bit regime.
>
> > Could the authors provide per-layer error curves or singular-value spectra to substantiate the claim that heavy-tailed distributions benefit most?
>
> To substantiate our claim quantitatively, we have added a **per-layer rate-distortion** analysis in *Figure 15*. The results demonstrate that NWC achieves lower MSE than baselines (e.g., QTIP), particularly in layers exhibiting high kurtosis (i.g., heavy-tailed) distributions such as Q/K projection layer of the first block.
>
> This improvement stems from the NWC encoder's ability to **suppress outliers**: it transforms heavy-tailed weights far more amenable to entropy coding than fixed basis transformations, which we discussed in **Appendix D.3**.
>
> ----
>
> ### **W3Q2.** Explicit decoding time comparison against baselines
> > Table 2 shows that decoding is dominated by entropy-decoding cost (~14 ms per 256×256 tensor on RTX 6000). Since the method requires an additional decoding network, the real deployment gains relative to lightweight quantizers such as GPTQ [1] or QuIP [3] are unclear. A direct wall-clock comparison or FLOP breakdown would be valuable.
>
> > What is the total encoding/decoding time compared to GPTQ [1] or QuIP [3] on the same GPU?
>
> Following your suggestion, we have updated **Table 3** to include the wall-clock encoding and decoding time comparisons with the baselines (GPTQ, QuIP#, QTIP). Note that we have also updated the latencies of the proposed NWC to a version where we use GPU-based entropy encoding/decoding, which has reduced the latencies to roughly 1/20. From the table, we observe that while NWC is slower than baselines, the gap is not excessively large; comparing with the QuIP#, NWC introduces ~30% overhead.
>
> Finally, we note that the decoding time may not be critical for many potential applications of NWC, e.g., bandwidth reduction in inter-chip communication, distributed training relying on periodic weight exchange, and storage optimization for massive personalized weights.

---

> ### Author Response · Authors · 2025-11-26
>
> ### **W4.** Clarifications on Fine-tuning Cost (Compute and Data).
> > The codec must be fine-tuned for each architecture. It is unclear how much data and computing are needed for these adaptations. Reporting training cost or sample requirements would help assess scalability.
>
> We have added a detailed breakdown of fine-tuning costs in **Appendix A.3** of the revised manuscript.
>
> **Compute cost**: Fine-tuning the codec for a 8B-scale architecture requires only ~1 hour on a single NVIDIA A6000 Ada GPU. The computational overhead is minimal; even for a massive 400B-scale model, fine-tuning is estimated to complete within 7 hours on a standard 8-GPU node (in practice, we anticipate significantly earlier completion due to rapid convergence).
>
> **Data requirements**: The codec training does not require any external datasets or text corpora, and uses only the model weights, effectively eliminating any overhead associated with data acquisition.
>
> ----
>
> ### **W5.** Additional experiments (other model architectures)
> > The main experiments use LLaMA-2 and LLaMA-3, but newer models such as LLaMA-4 and Qwen-3 are already available and would provide a stronger test of generality.
>
> According to the suggestion, we have appended additional experimental results on **Qwen3-30B-A3B** in Figure 6 of the revised manuscript. The NWC performance is strong at 4-6bits, similarly to Llama, demonstrating the broad architectural generality of our codec.
>
> Moreover, we are currently conducting additional experiments on GPT-OSS, as it is considered a more performant baseline (than Llama 4). We will add these results as soon as they become available.
>
> ----
> ### **Q3.**  Clarifications on generalizability of codec without retraining
> > Does the codec trained on one model (e.g., LLaMA-3-8B) generalize to another without retraining?
>
> As demonstrated in **Figures 5 and 11** of the original manuscript, the codec trained on Llama-3-8B generalizes well to other Llama models (Llama-2-7B and 13B).
>
> Furthermore, the additional experiments on Qwen3 and Mixtral presented in **Figure 6** confirm that the codec maintains its efficacy even on MoE-based models without retraining.
>
> ----
> ### **Q5.** Future direction: Hybrid scheme with analytical quantizers
> > Have you considered hybrid schemes combining NWC at mid-bitrates with analytical quantizers at extreme compression levels?
>
> Thank you for this insightful suggestion. While the idea sounds very interesting, we may need a “progressive compression” for such a hybrid approach to be meaningful in terms of the storage cost—i.e., we need to be able to reconstruct both mid-bit and low-bit versions of the model from the same code. The neural compression approach opens up the possibility for such progressive compression [1, 2, 3], but we expect significant additional effort should be made to bring such ideas into action.
>
>
> ----
> [1] Jeon et al., Context-based trit-plane coding for progressive image compression, CVPR 2023.
>
> [2] Lu et al., Progressive neural image compression with nested quantization and latent ordering, 2021.
>
> [3] Lee et al., DPICT: Deep progressive image compression using trit-planes, CVPR 2022.

---

### Author Response · Authors · 2025-11-26
**General Response**

Dear reviewers and Area Chairs,

We are truly grateful for many insightful and thoughtful comments from all the reviewers.

We are encouraged to find that the reviewers recognized our work addresses **a timely and well-motivated** problem (L5q5, H8K4, CDm5) with a **clear presentation** (L5q5, H8K4, XFXJ, CDm5), **supported by strong empirical results** (L5q5, XFXJ).

***
Through individual responses and revised manuscript, we have carefully addressed the questions and concerns of each reviewer. For your convenience, we provide a short summary of additional experiments, analyses, and clarifications.

**Experiments and Analyses**

- Additional experiment on other model architectures (Figure 6)
  - Qwen 3
  - Mixtral
- Additional baseline comparison (Figure 12)
  - Neural codec baseline
  - QAT and SVD baselines
- Decoding latency analysis (Table 3)
  - GPU-based entropy decoding
  - Comparison with GPTQ/QuIP#/QTIP
- Multi-seed variance analysis (Response to H8K4)
- Additional analyses
  - Per-layer rate-distortion (Section D.2)
  - Learned function analysis (Section D.3)
- Additional ablation studies
  - Ablation on learned networks (Table 4)
  - Ablation on entropy coding (Figure 13)
  - Ablation on normalization granularity (Figure 9)

**Clarifications**

- Difference between the proposed method and AWQ/GPTQ (Section E)
- Codec training & fine-tuning cost (Section A.3)
- Difference between the proposed framework and AQLM/LCQ/QuIP# (Section 2)
- Theoretical justification for Hessian diagonal approximation (Section F)
- Panter-Dite condition (Section 3.2)
- Gradient estimation (Response to CDm5)
- Notational clarity (Figure 3)
- Entropy model/coding details  (Section 4.2)
***
In the revised manuscript, the updates are temporarily highlighted in "red" for your convenience to check.

We strongly believe that the updates have greatly enhanced the quality of our manuscript.

Sincerely,
Authors

---

### Meta-Review · Area_Chair_uStu · 2026-01-08

**Summary:**

Reviewers note that decoding overhead makes the method impractical for deployment, similar approaches exist (AQLM, LCQ, QuIP#), and improvements are marginal. Authors provided extensive rebuttals with additional experiments on Qwen3/Mixtral, neural codec baselines, and GPU-based decoding latency analysis showing improved practicality. Despite strong engagement, reviewers maintained concerns about the fundamental deployment overhead. I recommend rejection.

**Reviewer Concerns:**

see above

**Reviewer Scores:**

Discussion quality was great; authors engaged thoroughly with detailed experiments addressing most concerns, but scores would have remained similar as core practicality issues persist.

---

### Decision · Program_Chairs · 2026-01-26

Reject